# Pan-Canadian survey on the impact of the COVID-19 pandemic on cervical cancer screening and management: cross-sectional survey of healthcare professionals

**Mariam El-Zein\*, Rami Ali, Eliya Farah, Sarah Botting-Provost, Eduardo L Franco, Survey Study Group**

Division of Cancer Epidemiology, McGill University, Montreal, Canada

## Abstract

**Background:** The coronavirus disease 2019 (COVID-19) pandemic has caused disruptions to cancer care by delaying diagnoses and treatment, presenting challenges and uncertainties for both patients and physicians. We conducted a nationwide online survey to investigate the effects of the pandemic and capture modifications, prompted by pandemic-related control measures, on cervical cancer screening-related activities from mid-March to mid-August 2020, across Canada.

**Methods:** The survey consisted of 61 questions related to the continuum of care in cervical cancer screening and treatment: appointment scheduling, tests, colposcopy, follow-up, treatment of pre-cancerous lesions/cancer, and telemedicine. We piloted the survey with 21 Canadian experts in cervical cancer prevention and care. We partnered with the Society of Canadian Colposcopists, Society of Gynecologic Oncology of Canada, Canadian Association of Pathologists, and Society of Obstetricians and Gynecologists of Canada, which distributed the survey to their members via email. We reached out to family physicians and nurse practitioners via MDBriefCase. The survey was also posted on McGill Channels (Department of Family Medicine News and Events) and social media platforms. The data were analyzed descriptively.

**Results:** Unique responses were collected from 510 participants (November 16, 2020, to February 28, 2021), representing 418 fully and 92 partially completed surveys. Responses were from Ontario (41.0%), British Columbia (21.0%), and Alberta (12.8%), and mostly comprised family physicians/general practitioners (43.7%), and gynecologist/obstetrician professionals (21.6%). Cancelled screening appointments were mainly reported by family physicians/general practitioners (28.3%), followed by gynecologist/obstetrician professionals (19.8%), and primarily occurred in private clinics (30.5%). Decreases in the number of screening Pap tests and colposcopy procedures were consistently observed across Canadian provinces. About 90% reported that their practice/institution adopted telemedicine to communicate with patients.

**Conclusions:** The area most severely impacted by the pandemic was appointment scheduling, with an important level of cancellations reported. Survey results may inform resumptions of various fronts in cervical cancer screening and management.

**Funding:** The present work was supported by the Canadian Institutes of Health Research (operating grant COVID-19 May 2020 Rapid Research Funding Opportunity VR5-172666 Rapid Research competition and foundation grant 143347 to Eduardo L Franco). Eliya Farah and Rami Ali each received an MSc stipend from the Department of Oncology, McGill University.

**\*For correspondence:**
mariam.elzein@mcgill.ca

**Group author details:**
Survey Study Group See page 27

## Editor's evaluation

This study explored practitioners' assessments of the impact of the pandemic on cervical cancer screening and follow-up. This is a very important topic that could continue to have implications for how this screening process is delivered now, after the pandemic.

## Introduction

Following the announcement of the coronavirus disease 2019 (COVID-19) pandemic by the World Health Organization in mid-March 2020 (*Ghebreyesus, 2020*), the severe acute respiratory syndrome coronavirus 2 (SARS-CoV-2) has since spread across the globe, resulting in enough severe illness to overwhelm the healthcare system (*Blumenthal et al., 2020*; *McMahon et al., 2020*). The immediate, preventive measures taken in response have adversely affected an entire range of activities, specifically those related to cancer control, prevention, and care (*Farah et al., 2021*). As a result, cancer screening and treatment services have been scaled back to conserve resources, increase capacity for managing COVID-19 patients, and lower the risk of infections among cancer patients worldwide, particularly during full and partial lockdown periods. Elective surgery, chemotherapy, and radiotherapy procedures have been postponed or modified, which necessitated rapid adaptation by the medical community and adjustment in health services, including the use of telehealth. About one-third of family medicine physicians in North America reported delaying cancer screening in the early phase of the pandemic and while some physicians reported high use of telehealth, most reported that reductions in cancer screenings would lead to increased incidence of late-stage cancers (*Price et al., 2022*).

Significant decreases in cancer diagnoses have been reported in multiple affected countries (*Dinmohamed et al., 2020*; *Cancer Australia, 2020*), alongside challenges in delivering timely cancer care to patients (*Gourd, 2021*; *Wilkinson, 2020*; *Maringe et al., 2020*; *Chen-See, 2020*). According to a survey carried out between May 22 and June 10, 2020, by the Canadian Cancer Survivor Network, 54% of patients reported that their cancer care-related appointments were cancelled, postponed, or rescheduled due to the pandemic, and 71% expressed concerns about their ability to receive proper care, testing, and follow-up appointments in a timely fashion (*Canadian Cancer Survivor Network, 2020*). A stochastic microsimulation model using data from the Canadian Cancer Registry predicted that pandemic-related cancer care disruptions (March 2020 to June 2021) could lead to 21,247 (2.0%) more cancer deaths in Canada between 2020 and 2030, if treatment capacity in 2021 were to recover to 2019 pre-pandemic levels (*Malagón et al., 2022*).

Breast and cervical cancer screening tests in the United States declined by 87% and 84%, respectively, during April 2020 compared with the previous 5-year averages for the month of April (*DeGroff et al., 2021*). In Ontario, Canada, there were 41% fewer screening tests delivered in 2020 for breast, cervical, colorectal, and lung cancer than in 2019 (*Walker et al., 2021*). A population-based study in Ontario, Canada, found that, between March and August 2020, the average monthly number of cytology tests, colposcopies, and treatments decreased by 63.8%, 39.7%, and 31.1%, respectively, compared to the same months in 2019 and that on average there were 292 fewer high-grade cytological abnormalities (decrease by 51.0%) detected each month (*Meggetto et al., 2021*).

We conducted a national cross-sectional survey-based descriptive study among healthcare professionals to (1) assess and portray the early impact of the pandemic on cervical cancer screening, diagnosis, management, and treatment services across Canada, (2) identify actionable approaches used by experts to mitigate the impact of the pandemic on their practice, and (3) identify windows of opportunity that were created by the pandemic and pinpoint positive aspects that could potentially enable the transformation of cervical cancer screening.

## Materials and methods

### Target population

The survey questions were formulated to gather the opinions and firsthand experiences of colposcopists, colposcopy registered nurses, registered practical nurses, cytopathologists, technologists, general practitioners, family physicians, obstetrician and gynecology staff, gynecological oncologists, gynecology nurses, pathologists, and physician assistants working in private and public health institutions in Canada.

**eLife digest** Cervical cancer is a common cancer among women caused by infections with certain types of human papillomavirus (HPV). Nearly four in five people are infected with HPV during their lifetime, making it the most common sexually transmitted infection worldwide. Vaccination against the virus can prevent infections and routine screening for precancerous lesions can enable early diagnosis and treatment, improving outcomes.

However, the COVID-19 pandemic has disrupted routine cervical cancer screening programs in several countries. This has caused delays in screening, which could result in more women being diagnosed with advanced-stage cancers.

El-Zein et al. showed that despite the interrupted screening programmes, about half of practices in Canada were able to catch up on delayed screening by February 2021. Between November 2020 and February 2021, El-Zein et al. surveyed 510 Canadian healthcare professionals involved in cervical cancer screening and treatment. About 64%-75% of the respondents reported canceled or postponed screening appointments. Most appointment delays were less than four months. Fewer than one in ten delays were longer than six months.

Most survey respondents said their practices pivoted to using telemedicine for some patient visits, such as cervical cancer screening follow-ups. About 40% of respondents suggested that the pandemic provided support to alternative screening options, such as HPV self-sampling at home. The survey results may help healthcare professionals and policymakers to develop plans that mitigate disruptions to cervical cancer screening during future emergencies.

## Survey design, development, and validation

We used the Checklist for Reporting Results of Internet E-Surveys (CHERRIES) to guide survey development and reporting of results (*Eysenbach, 2004*). The survey (*Supplementary file 1*), designed by members of the research team, consisted of 61 questions including informed e-consent and occupational demographics (questions Q2-Q5) such as attributes of the specialty, provider type, and affiliations of respondents. It was constructed around five themes related to screening practice (Q6-Q37), treatment of pre-cancerous lesions and cancer (Q38-Q42), telemedicine (Q43-Q47), over- and under-screening in the pre-COVID-19 era (Q48-Q51), and resumption of in-person practice (Q52-Q61). The first two themes covered a range of questions that reflected the continuum of care in cervical cancer screening and management. The screening practice theme included sub-sections focusing on appointment scheduling (Q6-Q13), screening tests (Q14-Q21), human papillomavirus (HPV) self-sampling (Q22-Q23), colposcopy (Q24-Q29), and screening follow-up (Q30-Q37). Questions 6 through 47 were designed to collect data during the early COVID-19 period spanning from mid-March until mid-August 2020. For questions pertaining to the 'resumption of in-person practice', the period of interest was from mid-August 2020 until the date of survey completion. We also collected data on sex and age of the respondents. Respondents were asked to provide their impressions and best estimates when completing the survey, without necessarily confirming the proportions that they reported with their institution's statistics.

For content validation and to determine question suitability and flow prior to the launch of the online survey, we conducted three iterative rounds of pilot testing by distributing the initial survey questionnaire to members of the Survey Study Group, consisting of 21 leading cervical cancer specialists and physicians in Canada who were not involved in study conception or design. Collective feedback in terms of relevance, appropriateness, and clarity of theme-related questions, as well as questionnaire length, was incorporated into the survey after each round; it was also used to refine the wording, type, and order of questions. Most were closed-ended (nominal, ordinal, and Likert-type questions), with few free-text questions and sub-questions that required elaborated responses.

## Survey administration and data management

The survey, constructed as a web-based questionnaire (originally developed in English and translated to French), was administered using LimeSurvey, an online-based survey tool hosted by McGill University. It was pretested by our research team and the panel of experts to ensure experiential functionality and valid data collection.

Several professional societies advertised and disseminated the survey via their email newsletter to respective members. These included the Society of Canadian Colposcopists (SCC; first email sent on November 18, 2020), Society of Gynecologic Oncology of Canada (GOC; first email sent on November 27, 2020), Canadian Association of Pathologists (CAP; first email sent on November 18, 2020), and Society of Obstetricians and Gynecologists of Canada (SOGC; first email sent on January 18, 2021). We reached out to primary care providers (i.e., family physicians, as well as general and nurse practitioners) via MDBriefCase (first email sent on February 7, 2021), which provides online continuing professional development to healthcare practitioners. Other platforms were also utilized to reach our targeted population, including posting a link to participate on the McGill Department of Family Medicine website, sending a request for participation letter containing the link to the listserv of the Chairs of Departments of Family Medicine across Canada, and social media platforms (LinkedIn and Twitter). The bilingual invitations to voluntary participate contained an email link to the web-based questionnaire survey. Two reminder emails – reiterating the objectives of the survey, inclusion criteria, and the survey link – were sent periodically (every 3–4 weeks).

Upon first entry to the survey portal, an informed e-consent form included a description of the survey, its objectives, and assurance of confidentiality of survey responses. Respondents who did not provide e-consent were unable to proceed to the survey questions. The platform allowed respondents to navigate between the different themes to revise their answers, if needed. Respondents received a $5 Starbucks gift card incentive upon completion of the survey, conditional on providing a valid email address.

The survey data, collected anonymously (no personal identifiers or IP addresses), were imported from LimeSurvey and curated in Excel (data cleaning and validation). We applied a two-step process to determine inclusion, the first based on eligibility criteria (target population) and the second on quality checks by flagging suspicious responses in open-ended questions. These questions entailed providing justification in Q22 and Q23 (if yes, no, or maybe, briefly justify); specification in Q47, Q54 (if other, specify), and Q61; description in Q60 (if yes, briefly describe); and any additional comments at the end of the survey.

Analyses included descriptive statistics and summaries of responses by province/territory (Alberta, British Columbia, Ontario, all other provinces, and territories), profession (primary care [i.e., general practitioners, family physicians, nurse practitioners, and physician assistants]; secondary care involving clinical diagnosis [i.e., colposcopists and colposcopy registered nurses/register nurse practitioners]; secondary care involving cytopathological diagnosis [i.e., cytopathologists and pathologists]; and tertiary care involving gynecology and related activities [i.e., gynecological oncologists, gynecologists, obstetrician-gynecologists, and gynecology nurses]), and place of practice (university-affiliated hospital, community hospital, public clinic, private clinic, community health center, and other). When responses to a given question were incomplete, we used the total number of complete responses as the denominator. Open-ended questions were analyzed using content analysis. Excel and SAS version 9.4 were used for data cleaning/visualization and analysis, respectively.

## Results

### Survey administration and responses

As shown in *Figure 1*, 778 potentially eligible respondents clicked on the survey link. Of those who started the survey, 235 were excluded as they were non-Canadian, had non-valid professions or places of practice, left the survey blank, or only completed the demographic section. Another 33 surveys were considered questionable; respondents took the survey multiple times, gave multiple non-sequitur or contradictory answers, or plagiarized responses (copied text from websites/Internet Google search). The final analysis sample comprised answers from 510 individuals, among whom the median time spent to complete the survey was 11 min and 53 s (interquartile range 6 min and 52 s to 17 min and 46 s). The survey was completed between November 16, 2020, and February 28, 2021.

### Characteristics of survey respondents

*Table 1* summarizes the characteristics of the study population. There were more female than male respondents. The mean age was 44.4 years±11.9 (range 20–86, median: 42 years, interquartile range 35–54). Responses were mainly from Ontario, followed by British Columbia and Alberta. Most respondents were general practitioners/family physicians (43.7%), gynecologists/obstetrician-gynecologists (21.6%), nurse practitioners/registered nurses (14.1%), and colposcopists (10.2%). Regarding the place of

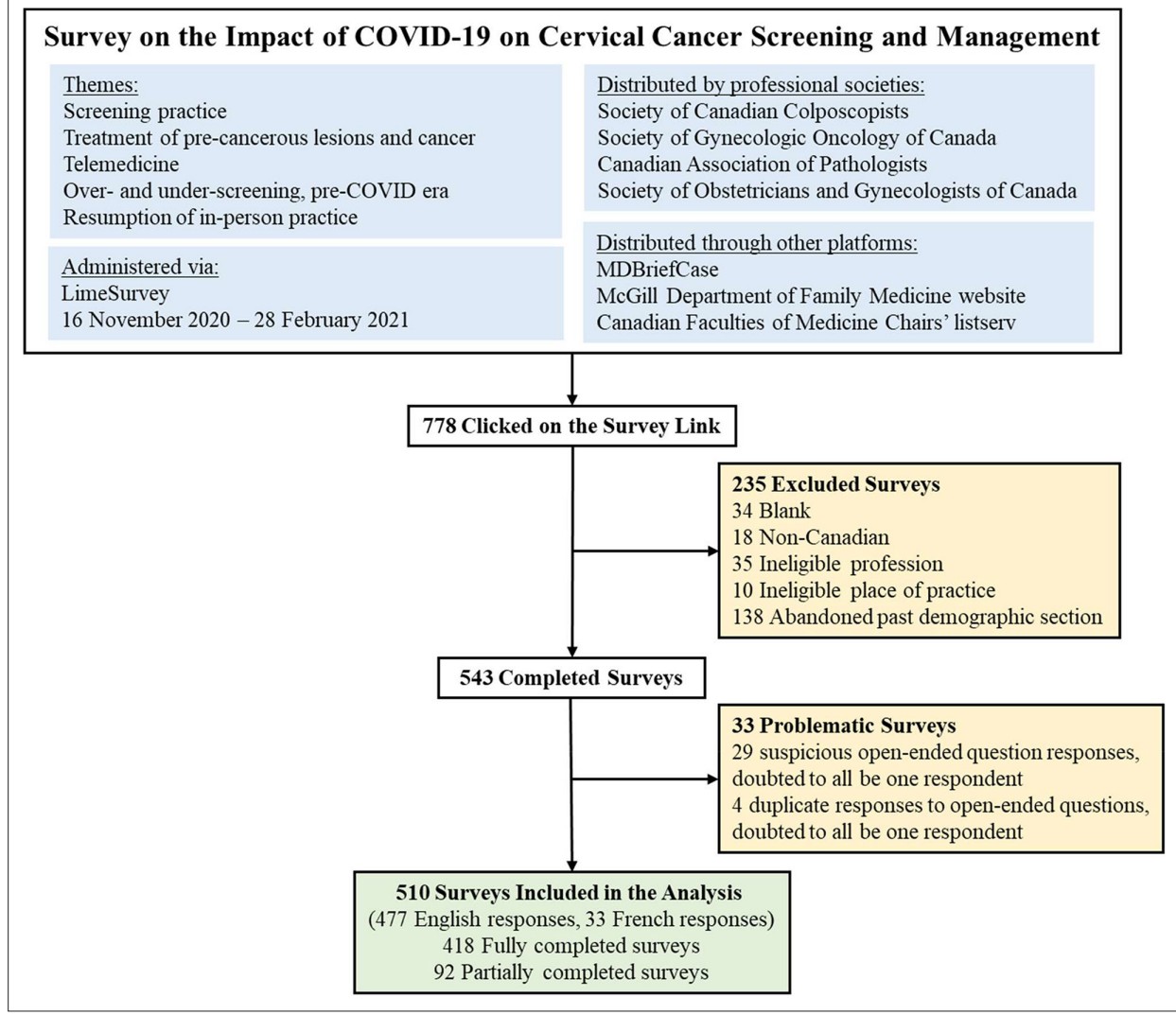

**Figure 1.** Description of survey elements and administration, and respondent flowchart.

practice, 32.9% reported working in private clinics, whereas comparable proportions reported working in university-affiliated hospitals (24.3%), community-affiliated hospitals (27.8%), and public clinics (25.3%). Some respondents selected multiple professions and/or places of practice. Of note, 42 of the 52 colposcopists were also gynecologists/obstetrician-gynecologists. Of the 124 respondents who practice in a university-affiliated hospital, 14, 11, and 16 respondents also selected community hospital, public clinic, and private clinic as a place of practice, respectively. Additionally, of the 142 respondents who work in a community hospital, 17 stated practicing in a public clinic and 23 in a private clinic.

## Theme 1: Screening practice

Cancellations and postponements of screening appointments were reported by 63.7% and 74.9% of respondents, respectively (*Table 2*). These are characterized in *Figure 2* by province (largely reported by healthcare professionals in Ontario), profession (largely reported by those in primary settings), and place of practice (largely reported by those in private clinics). Of the 325 respondents who reported cancellations of appointments, 55.7% stated that up to 49% of these appointments were cancelled by the physician or provider's institution (*Table 2*). Similarly, 63.7% and 40.6% reported that up to 49% were cancelled by the patient or converted to telemedicine, respectively. Of the 382 healthcare professionals who reported that appointments were postponed, 51.6%, 68.4%, and 42.9% respectively stated that up to 49% of these appointments were postponed by the physician or provider's institution, by the patient, or converted to telemedicine. The majority of appointments (64.4%)

**Table 1.** Characteristics of survey respondents (n=510).

| Variable | Categories | n (%) |
|---|---|---|
| | Female | 284 (55.7) |
| | Male | 124 (24.3) |
| Sex | *Not reported* | *102 (20.0)* |
| | 20–29 | 26 (5.1) |
| | 30–39 | 122 (23.9) |
| | 40–49 | 98 (19.2) |
| | 50–59 | 70 (13.7) |
| | 60–69 | 42 (8.2) |
| | 70+ | 7 (1.4) |
| Age | *Not reported* | *145 (28.4)* |
| | Alberta | 65 (12.8) |
| | British Columbia | 107 (21.0) |
| | Manitoba | 18 (3.5) |
| | New Brunswick | 19 (3.7) |
| | Newfoundland and Labrador | 7 (1.4) |
| | Northwest Territories | 9 (1.8) |
| | Nova Scotia | 21 (4.1) |
| | Nunavut | 4 (0.8) |
| | Ontario | 209 (41.0) |
| | Prince Edward Island | 2 (0.4) |
| | Quebec | 21 (5.1) |
| | Saskatchewan | 26 (5.1) |
| | Yukon | 1 (0.2) |
| Province/territory | *Not reported* | *1 (0.2)* |
| | Colposcopist | 52 (10.2) |
| | Colposcopy registered nurse/registered practical nurse | 16 (3.1) |
| | Cytopathologist/technologist | 44 (8.6) |
| | General practitioner/family physician | 223 (43.7) |
| | Gynecologist/obstetrician-gynecologist | 110 (21.6) |
| | Gynecology oncologist | 32 (6.3) |
| | Gynecology nurse | 21 (4.1) |
| | Nurse practitioner/registered nurse | 72 (14.1) |
| | Pathologist | 17 (3.3) |
| | Physician assistant | 7 (1.4) |
| Profession* | Other (manager in a community health center) | 1 (0.2) |
| | University-affiliated hospital | 124 (24.3) |
| | Community hospital | 142 (27.8) |
| | Public clinic | 129 (25.3) |
| | Private clinic | 168 (32.9) |
| | Community health center | 37 (7.3) |
| Place of practice* | Other (homeless shelter [nurse]; private lab [cytotechnologist]) | 2 (0.4) |

*Frequency count exceeded number of respondents (510) as some selected more than one answer.

**Table 2.** Cancellations and postponements of cervical cancer screening appointments.

| Question number and content (number of responses) | | Categories | n (%) |
|---|---|---|---|
| Q6 **Cancellations** of **screening** appointments (n=510) | | Yes | 325 (63.7) |
| | | No | 114 (22.4) |
| | | *Don't know* | *39 (7.7)* |
| | | *Not applicable to my practice* | *32 (6.3)* |
| Q7 Percentage of **cancelled screening** appointments (n=325) | Cancelled by **physician or provider's institution** | 0% | 37 (11.4) |
| | | 1–24% | 107 (32.9) |
| | | 25–49% | 74 (22.8) |
| | | 50–74% | 53 (16.3) |
| | | ≥75% | 35 (10.8) |
| | | *Don't know* | *19 (5.8)* |
| | Cancelled by **patient** | 0% | 8 (2.5) |
| | | 1–24% | 124 (38.2) |
| | | 25–49% | 83 (25.5) |
| | | 50–74% | 44 (13.5) |
| | | ≥75% | 42 (12.9) |
| | | *Don't know* | *24 (7.4)* |
| | **Converted to telemedicine** | 0% | 77 (23.7) |
| | | 1–24% | 88 (27.1) |
| | | 25–49% | 44 (13.5) |
| | | 50–74% | 56 (17.2) |
| | | ≥75% | 36 (11.1) |
| | | *Don't know* | *24 (7.4)* |
| Q8 **Postponements of screening** practices (n=510) | | Yes | 382 (74.9) |
| | | No | 73 (14.3) |
| | | *Don't know* | *24 (4.7)* |
| | | *Not applicable to my practice* | *31 (6.1)* |

*Table 2 continued on next page*

*Table 2 continued*

| Question number and content (number of responses) | | Categories | n (%) |
|---|---|---|---|
| | | 0% | 40 (10.5) |
| | | 1–24% | 110 (28.8) |
| | | 25–49% | 87 (22.8) |
| | | 50–74% | 66 (17.3) |
| | Postponed by **physician or provider's institution** | ≥75% | 54 (14.1) |
| | | *Don't know* | *25 (6.6)* |
| | | 0% | 12 (3.1) |
| | | 1–24% | 153 (40.1) |
| | | 25–49% | 108 (28.3) |
| | | 50–74% | 46 (12.0) |
| | | ≥75% | 33 (8.7) |
| | Postponed by **patient** | *Don't know* | *30 (7.9)* |
| | | 0% | 103 (27.0) |
| | | 1–24% | 104 (27.2) |
| | | 25–49% | 60 (15.7) |
| | | 50–74% | 53 (13.9) |
| Q9 Percentage of **postponed screening** appointments (n=382) | | ≥75% | 29 (7.6) |
| | **Converted to telemedicine** | *Don't know* | *33 (8.6)* |
| | | 1 week to <2 weeks | 15 (3.9) |
| | | 2 weeks to <4 weeks | 47 (12.3) |
| | | 1 month to <2 months | 66 (17.3) |
| | | 2 months to <4 months | 118 (30.9) |
| | | 4 months to <6 months | 78 (20.4) |
| | | >6 months | 36 (9.4) |
| Q10 Length of **deferral period** for **postponed screening** appointments (n=382) | | *Don't know* | *22 (5.7)* |

were at most deferred by less than 4 months, whereas 9.4% were deferred by more than 6 months. *Figures 3–5* show the proportions of all cancelled and postponed screening appointments by province, profession, and place of practice, respectively; most responses were once again from professionals in Ontario, primary care, and private clinics. Of the respondents who experienced Pap test deferral periods of 2 months or longer, 38% (100/26) worked in private clinics. Those who practiced in community hospitals reported deferral periods of 2 months or more for HPV test (36.8% [42/114]) and HPV/Pap co-test (42.7% [53/124]) more frequently than other places of practice. A total of 99 respondents (19.4%) reported that their practice/institution did not allow in person consultation appointments during the pandemic's peak period (*Supplementary file 2a*). Of those who reported allowance of in-person consultations (378, 74.1%), most were from Ontario, in primary care, and practicing in private clinics (*Figure 6*).

In terms of the type of test usually employed for primary cervical cancer screening (*Supplementary file 2b*), 76.1% of respondents reported cytology, 32.8% the HPV test, and 25.4% reported using both. Compared to pre-COVID-19, 15.8%, 4.9%, and 3.8% reported a decrease by 75% or more in the number of Pap, HPV, and co-tests, respectively. Delays in scheduling of these tests were correspondingly reported by 56.9%, 22.6%, and 21.8% of respondents. Of the 469 healthcare professionals who reported cancellations of a scheduled screening test, 48.1%, 19.8%, and 17.3% stated that up to 49% of Pap, HPV, and co-tests were cancelled, whereas of the 468 professionals who reported postponements, the corresponding proportions were 46.8%, 22.3%, and 20.1% (*Supplementary file 2c*). Pap tests (56.5%), HPV tests (31.5%) and HPV/Pap co-tests (25.5%) were deferred by less than 4 months, at the most, whereas 10.1%, 6.6%, and 6.6% of these tests were deferred by more than 6 months,

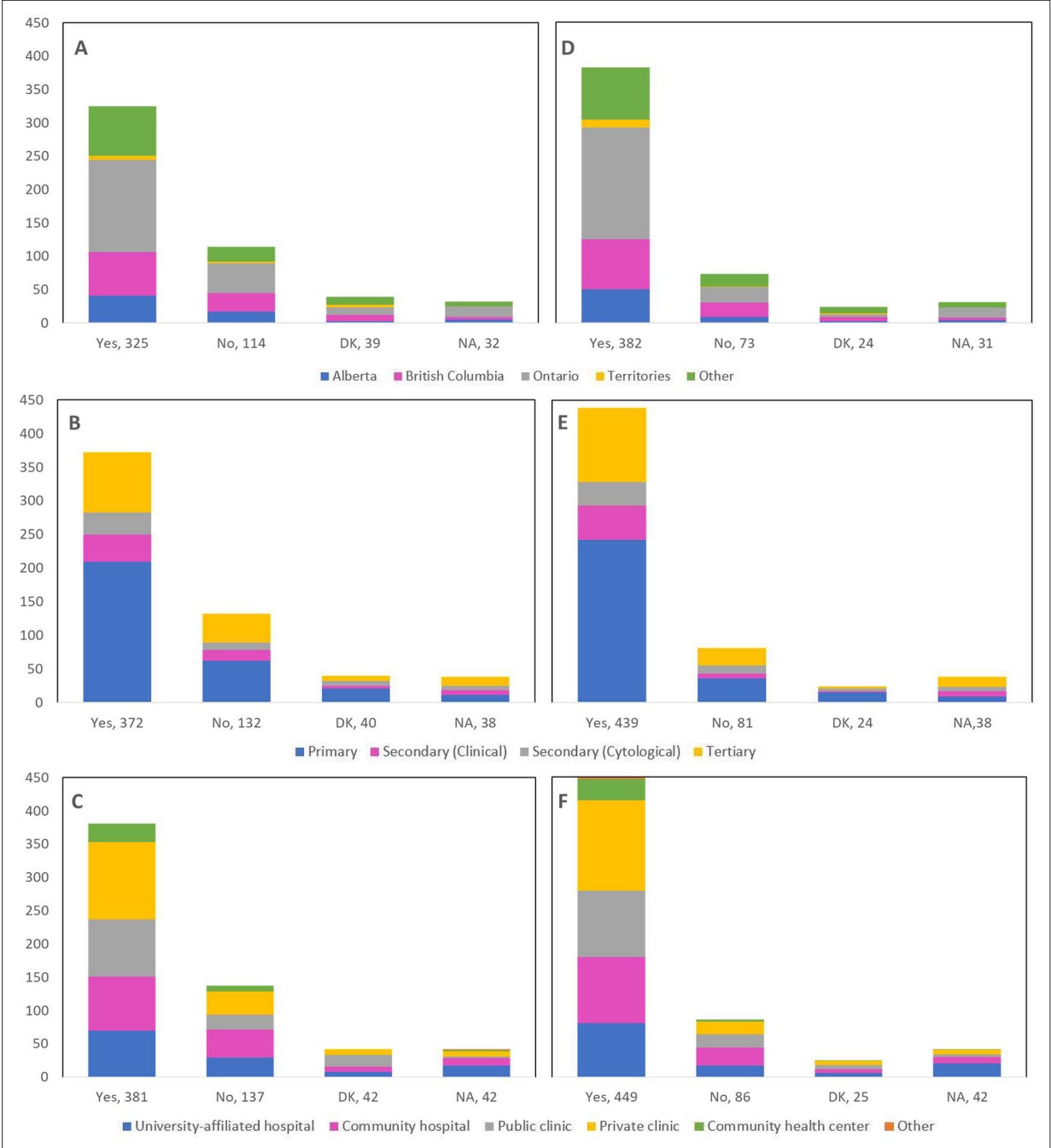

**Figure 2.** Cancellations and postponements of screening appointments by province, profession, and place of practice (n=510). Number of <u>cancellations</u> are shown by (**A**) province, (**B**) profession, and (**C**) place of practice. Number of <u>postponements</u> are shown by (**D**) province, (**E**) profession, and (**F**) place of practice. Answers include responses for questions 6 (cancellations) and 8 (postponements) by questions 2 (province), 4 (profession), and 5 (place of practice). Panels **A and D**: Territories include Northwest Territories, Nunavut, and Yukon. Other provinces include Manitoba, New Brunswick, Newfoundland and Labrador, Nova Scotia, Prince Edward Island, Quebec, and Saskatchewan (and one respondent who preferred not to say). Panels **B and E**: Primary includes general practitioners/family physicians, nurse practitioners/registered nurses, physician assistants, and a manager of a community health center; secondary (clinical) includes colposcopists and colposcopy registered nurses/registered practical nurses; secondary (cytological) includes cytopathologists/technologists and pathologists; tertiary includes gynecologists/obstetrician-gynecologists, gynecology oncologists, and gynecology nurses. Panels **B, C, E, and F**: Frequency count exceeded total number of respondents as some reported multiple professions and places of practice. DK: Don't know; NA: Not applicable to my practice.

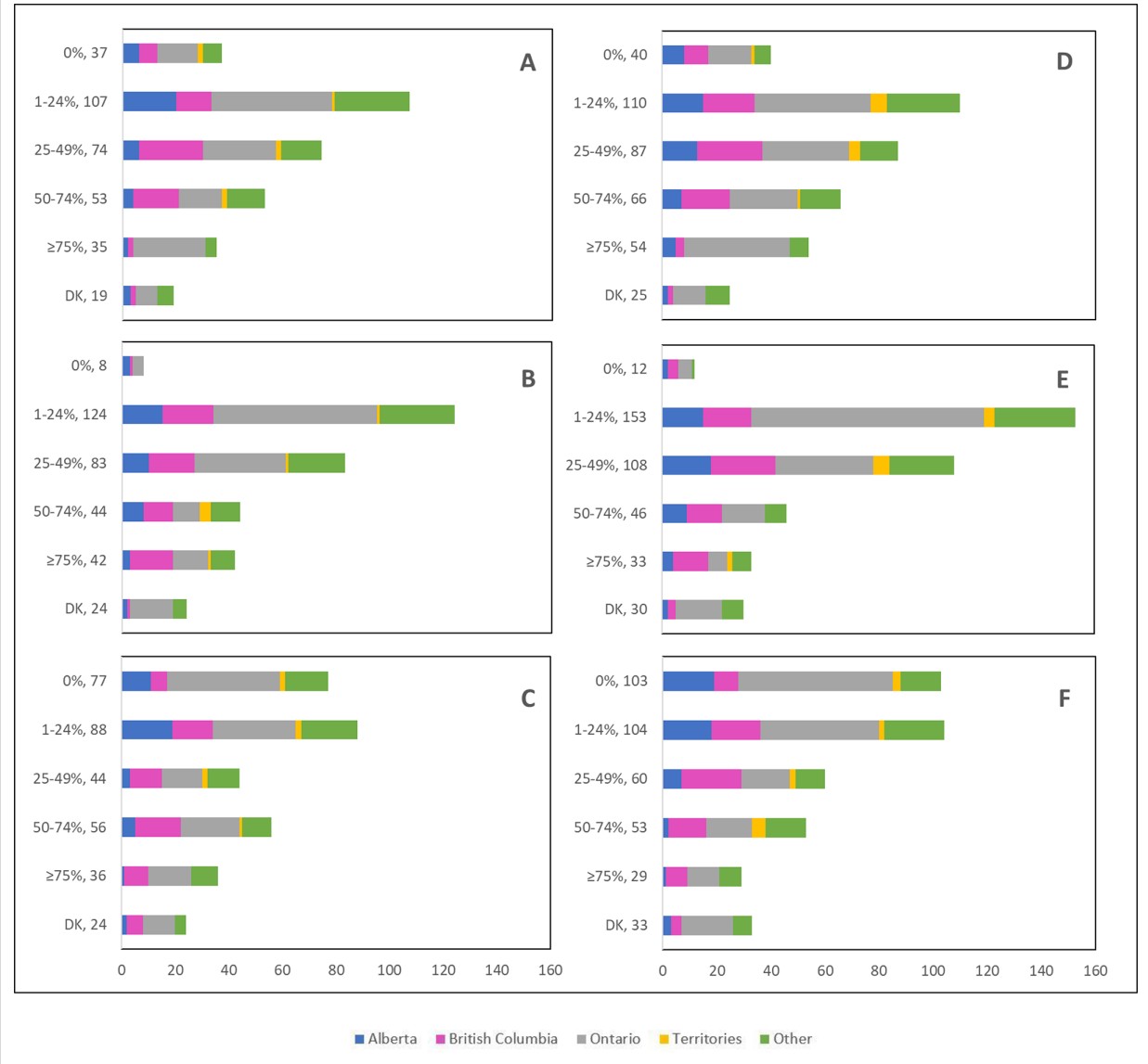

**Figure 3.** Cancelled (n=325) and postponed (n=382) screening appointments by province that were cancelled or postponed by physician/providers' institution, by patient, or converted to telemedicine. Number cancelled by (**A**) physician or providers' institution, (**B**) patient, and (**C**) <u>converted</u> to telemedicine. Number <u>postponed</u> by (**D**) physician or providers' institution, (**E**) patient, and (**F**) converted to telemedicine. Answers include responses for questions 7 (cancellations) and 9 (postponements) by question 2 (province). Respondents were asked to ensure that their answers did not exceed 100% for each question. (i.e., for each respondent, A+B+C ≈ 100% and D+E+F ≈ 100%). The x axis represents frequency of responses by province. Territories include Northwest Territories, Nunavut, and Yukon. Other provinces include Manitoba, New Brunswick, Newfoundland and Labrador, Nova Scotia, Prince Edward Island, Quebec, and Saskatchewan (and one respondent who preferred not to say). The y axis represents cancelled or postponed screening appointments using a predefined interval scale. DK: Don't know.

respectively. *Figure 7* illustrates the deferral period of these postponed screening tests appointments by province, profession, and place of practice. Regarding the delay in forwarding tests to the laboratory, 15%, 15.6%, and 13.9% of respondents reported such delays for Pap, HPV, and co-tests, respectively (*Supplementary file 2d*).

When asked about whether the pandemic will encourage/facilitate/accelerate the implementation of HPV self-sampling in cervical cancer screening programs at a provincial and/or national level, 150/455 respondents (33%) indicated that it will, and 50.1% were in favor of implementing this modality as an alternative screening method (*Supplementary file 2e*).

With respect to colposcopy appointments, cancellations, and postponements were reported by 25.2% and 37% of respondents (*Supplementary file 2f*), with patterns and changes by province,

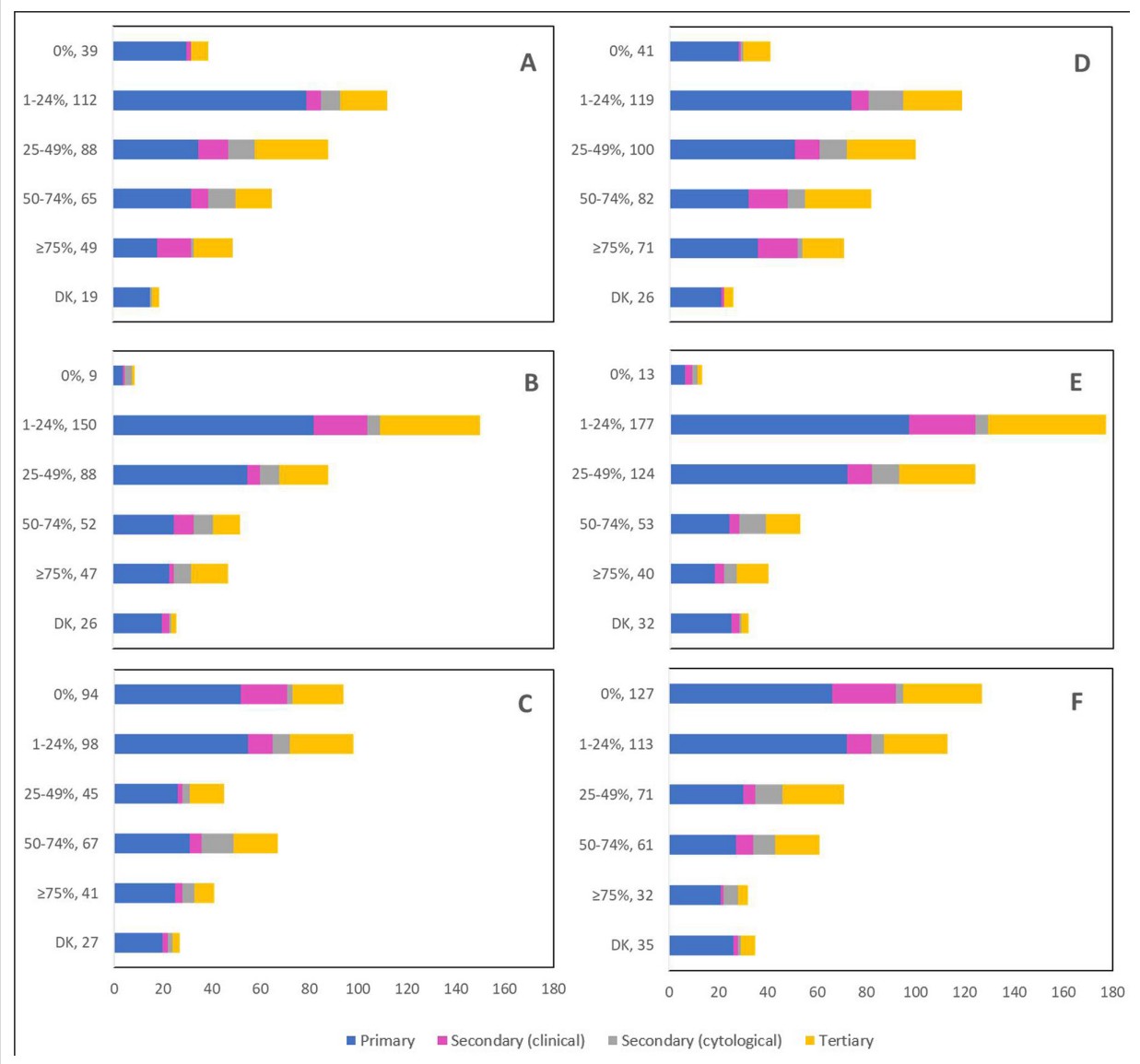

**Figure 4.** Cancelled (n=325) and postponed (n=382) screening appointments by profession that were cancelled or postponed by physician/providers' institution, by patient, or converted to telemedicine. Number cancelled by (**A**) physician or providers' institution, (**B**) patient, and (**C**) converted to telemedicine. Number postponed by (**D**) physician or providers' institution, (**E**) patient, and (**F**) converted to telemedicine. Answers include responses for questions 7 (cancellations) and 9 (postponements) by question 4 (profession). Respondents were asked to ensure that their answers did not exceed 100% for each question. (i.e., for each respondent, A+B+C ≈ 100% and D+E+F ≈ 100%). The x axis represents frequency of responses by profession. Primary includes general practitioners/family physicians, nurse practitioners/registered nurses, physician assistants, and a manager of a community hospital; secondary (clinical) includes colposcopists and colposcopy registered nurses/registered practical nurses; secondary (cytological) includes cytopathologists/technologists and pathologists; tertiary includes gynecologists/obstetrician-gynecologists, gynecology oncologists, and gynecology nurses. Frequency count exceeded total number of respondents as some reported multiple professions. The y axis represents cancelled or postponed screening appointments using a predefined interval scale. DK: Don't know.

profession, and place of practice (*Figure 8*, *Figure 8—figure supplements 1–3*) similar to those reported for screening appointments. The same was observed for reported cancellations (33.3%) and postponements (53.5%) of follow-up appointments (*Supplementary file 2g*, *Figure 9*, and *Figure 9—figure supplements 1–3*). Professionals from community hospitals saw longer deferral periods for postponed colposcopy appointments than those from other settings, accounting for 40.9% (38/93) of deferrals of 2 months or longer, whereas respondents from community health centers accounted for 34.9% (38/109) of deferrals of 2 months or longer of follow-up appointments. With respect to

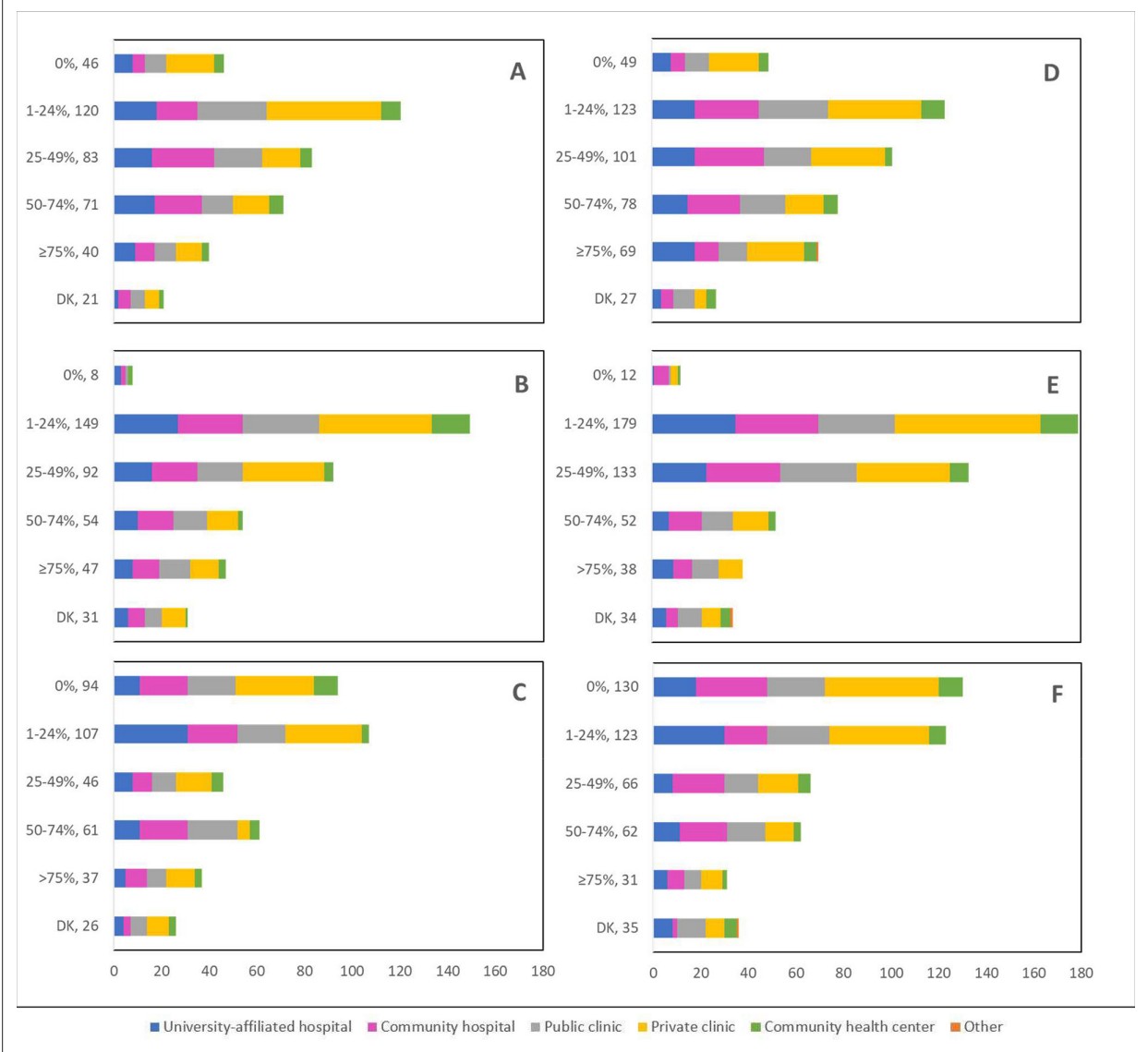

**Figure 5.** Cancelled (n=325) and postponed (n=382) screening appointments by place of practice that were cancelled or postponed by physician/providers' institution, by patient, or converted to telemedicine. Number cancelled by (**A**) physician or providers' institution, (**B**) patient, and (**C**) converted to telemedicine. Number postponed by (**D**) physician or providers' institution, (**E**) patient, and (**F**) converted to telemedicine. Answers include responses for questions 7 (cancellations) and 9 (postponements) by question 5 (place of practice). Respondents were asked to ensure that their answers did not exceed 100% for each question. (i.e., for each respondent, A+B+C ≈ 100% and D+E+F ≈ 100%). The x axis represents frequency of responses by place of practice. Frequency count exceeded total number of respondents as some reported multiple places of practice. The y axis represents cancelled or postponed screening appointments using a predefined interval scale. DK: Don't know.

receiving test results from the lab prior to follow-up with patients, 21.1%, 19.3%, and 12.1% of respondents reported delays for Pap, HPV, and co-tests, respectively (*Supplementary file 2h*).

## Theme 2: Treatment of pre-cancerous lesions and cancer

*Supplementary file 2i* presents the observed changes in the number of treatment procedures by treatment type reported by 431 respondents; cold knife conization (15.8% decrease, 12.5% unaffected, 15.8% increase), other excisional (20.0% decrease, 12.8% unaffected, 15.5% increase), ablative procedures (15.8% decrease, 10.9% unaffected, 16.3% increase), hysterectomy (23.9% decrease, 9.3% unaffected, 13.4% increase), chemotherapy (10.9% decrease, 10.9% unaffected, 15.1% increase), and radiation (13.7% decrease, 7.9% unaffected, 14.0% increase). The number of cancellations or postponements of treatment procedures (13.5% for cold knife conization, 23.7% for other excisional, 19.5%

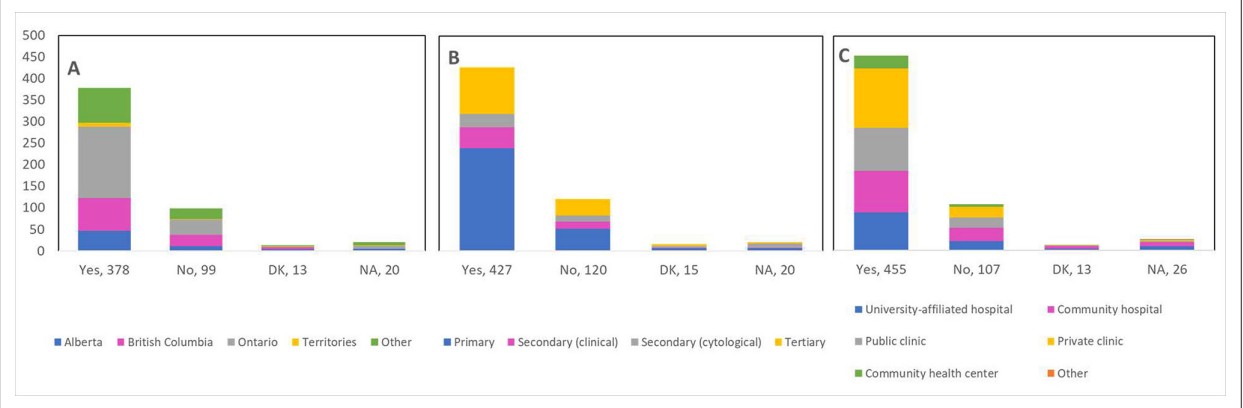

**Figure 6.** Allowance of in-person consultations during the peak period of the pandemic by province, profession, and place of practice (n=510). Number of in person consultations is shown by (**A**) province, (**B**) profession, and (**C**) place of practice. Answers include responses for question 11 by questions 2 (province), 4 (profession), and 5 (place of practice). Panel A: Territories include Northwest Territories, Nunavut, and Yukon. Other provinces include Manitoba, New Brunswick, Newfoundland and Labrador, Nova Scotia, Prince Edward Island, Quebec, and Saskatchewan (and one respondent who preferred not to say). Panel B: Primary includes general practitioners/family physicians, nurse practitioners/registered nurses, physician assistants, and a manager of a community health center; secondary (clinical) includes colposcopists and colposcopy registered nurses/registered practical nurses; Secondary (cytological) includes cytopathologists/technologists and pathologists; Tertiary includes gynecologists/obstetrician-gynecologists, gynecology oncologists, and gynecology nurses. Panels B and C: Frequency count exceeded total number of respondents as some reported multiple professions and places of practice. DK: Don't know; NA: Not applicable to my practice.

for ablative procedures, 21.1% for hysterectomy, 10.4% for chemotherapy, and 11.6% for radiation) are shown by province (*Figure 10*), profession (*Figure 11*), and place of practice (*Figure 12*). Community hospitals accounted for almost half of deferrals of 2 months or longer of cold knife conisation procedures (48.9% [43/88]), other excisional procedures (47.6% [39/82]), ablative procedures (46.7% [43/92]), hysterectomies (49.0% [47/96]), chemotherapy (46.8% [36/77]), and radiation (48.6% [35/72]).

## Theme 3: Telemedicine

*Table 3* presents responses reported by 429 respondents regarding the adoption of telemedicine. A total of 384 respondents (89.5%) reported that their practice/institution adopted telemedicine to communicate with patients; 26.8% indicated that they called 25–49% of their patients for distance consultations and 19.8% indicated the use of telemedicine with 25–49% of patients for follow-up appointments related to a cervical cancer screening procedure. Around two-thirds (72.7%) of healthcare professionals reported that virtual consultations are covered by their jurisdictional public health insurance system. Regarding which interactions with patients would be appropriate to convert to telemedicine, 82.1% of respondents selected test results reporting, 66% health and medical history reporting, 51.7% consent forms prior to in-person procedures, 42.9% post-procedure follow-up, and 33.1% selected in-person appointment planning/scheduling.

## Theme 4: Over- and under-screening in the pre-COVID-19 era

There was a total of 190 responses (44.5%) indicating issues of over-screening/over-diagnosis/over-treatment prior to the onset of the pandemic, with over-diagnosis (20.1%) of cervical lesions being the most commonly reported issue (*Table 4*). A minority of respondents reported that the current delays/cancellations of screening and management procedures may have had a positive impact in reducing unnecessary screening (15.2% of responses), diagnosis (20.6% of responses), and treatment (10.5% of responses). Conversely, 350 responses (81.9%) indicated issues of under-screening/under-diagnosis/under-treatment pre-COVID-19, and in turn, reported that the current delays/cancellations of screening and management procedures may have had a negative impact by reducing necessary screening (47.8% of responses), diagnosis (48.4% of responses), and treatment (25.3% of responses).

## Theme 5: Resumption of in-person practice

Nearly half (45.1%) of respondents reported that their practice/institution has caught up with the cancellations/postponements of appointments caused by restrictions introduced at the beginning of

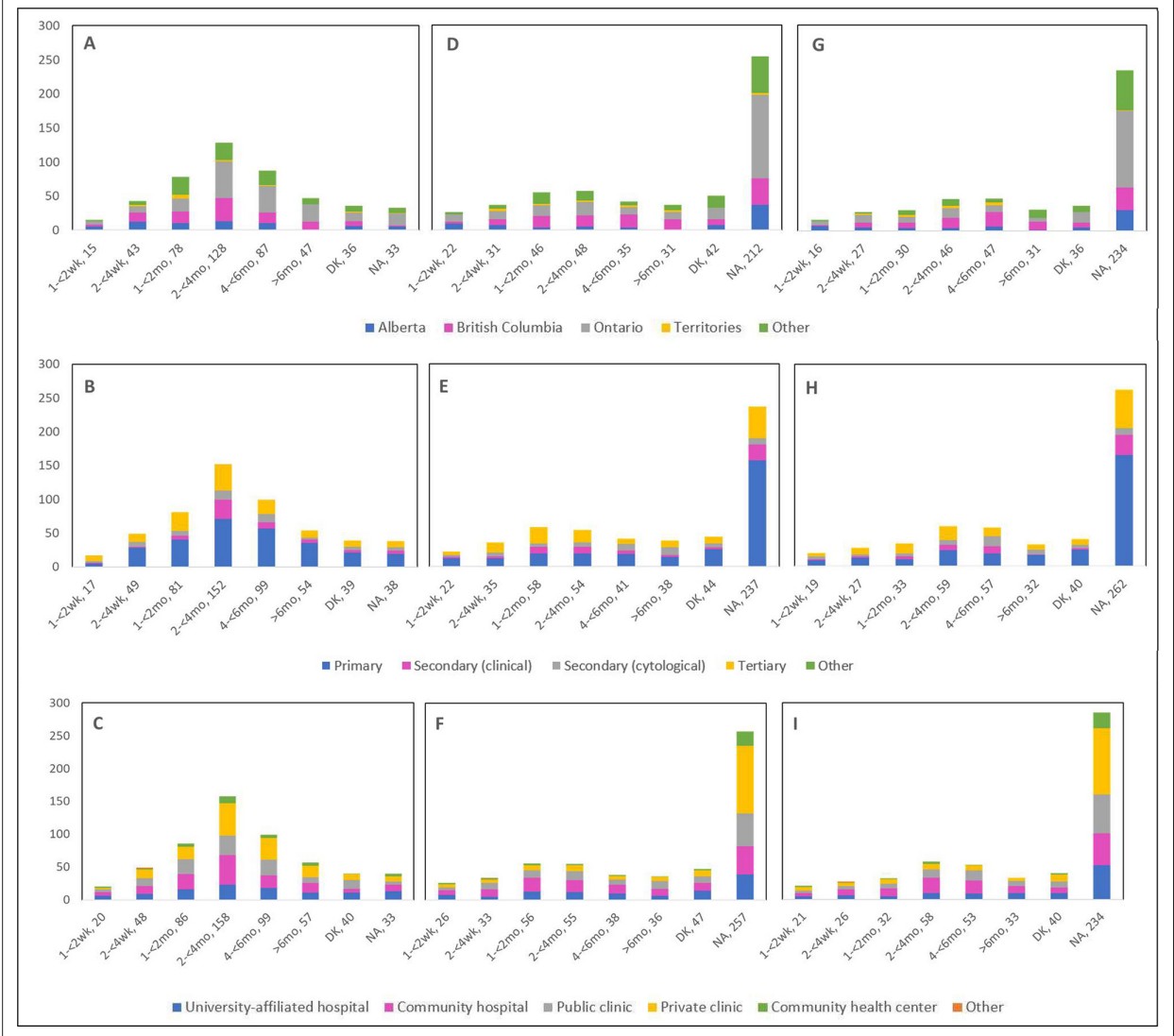

**Figure 7.** Deferral period for postponed screening appointments by province, profession, and place of practice (n=467). Deferral period for postponed Pap test appointments is shown by (**A**) province, (**B**) profession, and (**C**) place of practice. Deferral period for postponed human papillomavirus (HPV) test appointments is shown by (**D**) province, (**E**) profession, and (**F**) place of practice. Deferral period for postponed HPV/Pap co-test appointments is shown (**G**) province, (**H**) profession, and (**I**) place of practice. Answers include responses for question 19 by questions 2 (province), 4 (profession), and 5 (place of practice). Panels A, D, and G: Territories include Northwest Territories, Nunavut, and Yukon. Other provinces include Manitoba, New Brunswick, Newfoundland and Labrador, Nova Scotia, Prince Edward Island, Quebec, and Saskatchewan (and one respondent who preferred not to say). Panels B, E, and H: Primary includes general practitioners/family physicians, nurse practitioners/registered nurses, physician assistants, and a manager of a community health center; secondary (clinical) includes colposcopists and colposcopy registered nurses/registered practical nurses; secondary (cytological) includes cytopathologists/technologists and pathologists; tertiary includes gynecologists/obstetrician-gynecologists, gynecology oncologists, and gynecology nurses. Panels B, C, E, F, H, and I: Frequency count exceeded total number of respondents as some reported multiple professions and places of practice. DK: Don't know; NA: Not applicable to my practice.

the pandemic, whereas 34.9% reported ongoing disruptions and delays (*Table 5*). Allowing longer workdays and/or working on weekends, increasing availability of operating rooms for treatment procedures, and converting operating room procedures to take place in clinics constituted the main measures that were implemented to catch up with these cancellations/postponements. A total of 160 respondents (38%) indicated that their practice/institution has currently caught up with 50% or more of the cancellations/postponements. Nonetheless, 51.3% reported that patients have not been coming in for routine screening procedures at a capacity equivalent to the pre-COVID-19 era. Almost a third of respondents (29.2%) answered that 25–49% of patients have been attending routine screening

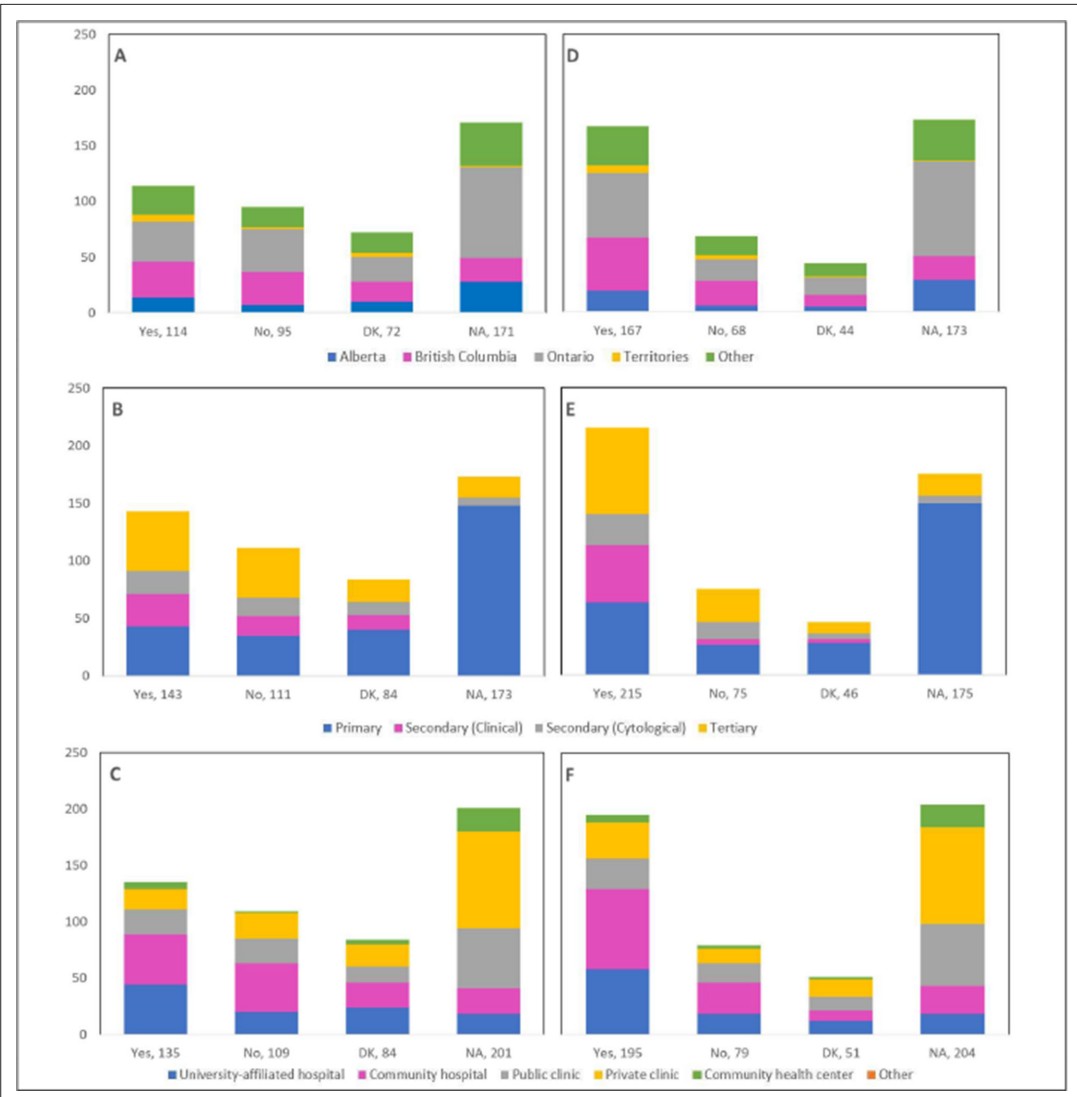

**Figure 8.** Cancellations and postponements of colposcopy appointments by province, profession, and place of practice (n=452). Number of cancellations are shown by (**A**) province, (**B**) profession, and (**C**) place of practice. Number of postponements are shown by (**D**) province, (**E**) profession, and (**F**) place of practice. Answers include responses for questions 25 (cancellations) and 27 (postponements) by questions 2 (province), 4 (profession), and 5 (place of practice). Panels A and D: Territories include Northwest Territories, Nunavut, and Yukon. Other provinces include Manitoba, New Brunswick, Newfoundland and Labrador, Nova Scotia, Prince Edward Island, Quebec, and Saskatchewan (and one respondent who preferred not to say). Panels B and E: Primary includes general practitioners/family physicians, nurse practitioners/registered nurses, physician assistants, and a manager of a community health center; secondary (clinical) includes colposcopists and colposcopy registered nurses/registered practical nurses; secondary (cytological) includes cytopathologists/technologists and pathologists; tertiary includes gynecologists/obstetrician-gynecologists, gynecology oncologists, and gynecology nurses. Panels B, C, E, and F: Frequency count exceeded total number of respondents as some reported multiple professions and places of practice. DK: Don't know; NA: Not applicable to my practice.

The online version of this article includes the following figure supplement(s) for figure 8:

**Figure supplement 1.** Cancelled (n=114) and postponed (n=167) colposcopy appointments by province that were cancelled or postponed by physician/providers' institution or by patient.

**Figure supplement 2.** Cancelled (n=114) and postponed (n=167) colposcopy appointments by profession that were cancelled or postponed by physician/providers' institution or by patient.

**Figure supplement 3.** Cancelled (n=114) and postponed (n=167) colposcopy appointments by place of practice that were cancelled or postponed by physician/providers' institution or by patient.

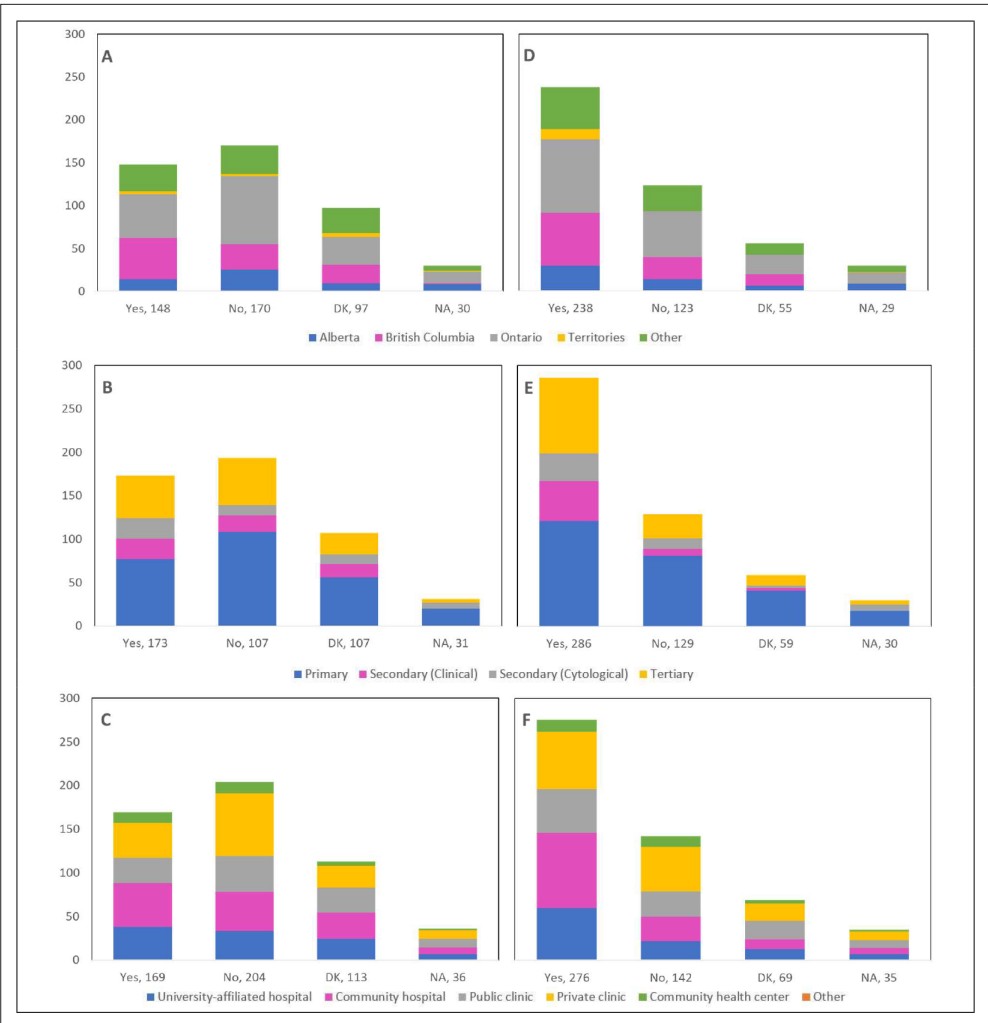

**Figure 9.** Cancellations and postponements of follow-up appointments by province, profession, and place of practice (n=445). Number of cancellations are shown by (**A**) province, (**B**) profession, and (**C**) place of practice. Number of postponements are shown by (**D**) province, (**E**) profession, and (**F**) place of practice. Answers include responses for questions 31 (cancellations) and 33 (postponements) by questions 2 (province), 4 (profession), and 5 (place of practice). Panels A and D: Territories include Northwest Territories, Nunavut, and Yukon. Other provinces include Manitoba, New Brunswick, Newfoundland and Labrador, Nova Scotia, Prince Edward Island, Quebec, and Saskatchewan (and one respondent who preferred not to say). Panels B and E: Primary includes general practitioners/family physicians, nurse practitioners/registered nurses, physician assistants, and a manager of a community health center; secondary (clinical) includes colposcopists and colposcopy registered nurses/registered practical nurses; secondary (cytological) includes cytopathologists/technologists and pathologists; tertiary includes gynecologists/obstetrician-gynecologists, gynecology oncologists, and gynecology nurses. Panels B, C, E, and F: Frequency count exceeded total number of respondents as some reported multiple professions and places of practice. DK: Don't know; NA: Not applicable to my practice.

The online version of this article includes the following figure supplement(s) for figure 9:

**Figure supplement 1.** Cancelled (n=148) and postponements (n=238) of follow-up appointments by province that were cancelled or postponed by physician/providers' institution, by patient, or converted to telemedicine.

**Figure supplement 2.** Cancelled (n=148) and postponed (n=238) follow-up appointments by profession that were cancelled or postponed by physician/providers' institution, by patient, or converted to telemedicine.

**Figure supplement 3.** Cancelled (n=148) and postponed (n=238) follow-up appointments by place of practice that were cancelled or postponed by physician/providers' institution, by patient, or converted to telemedicine.

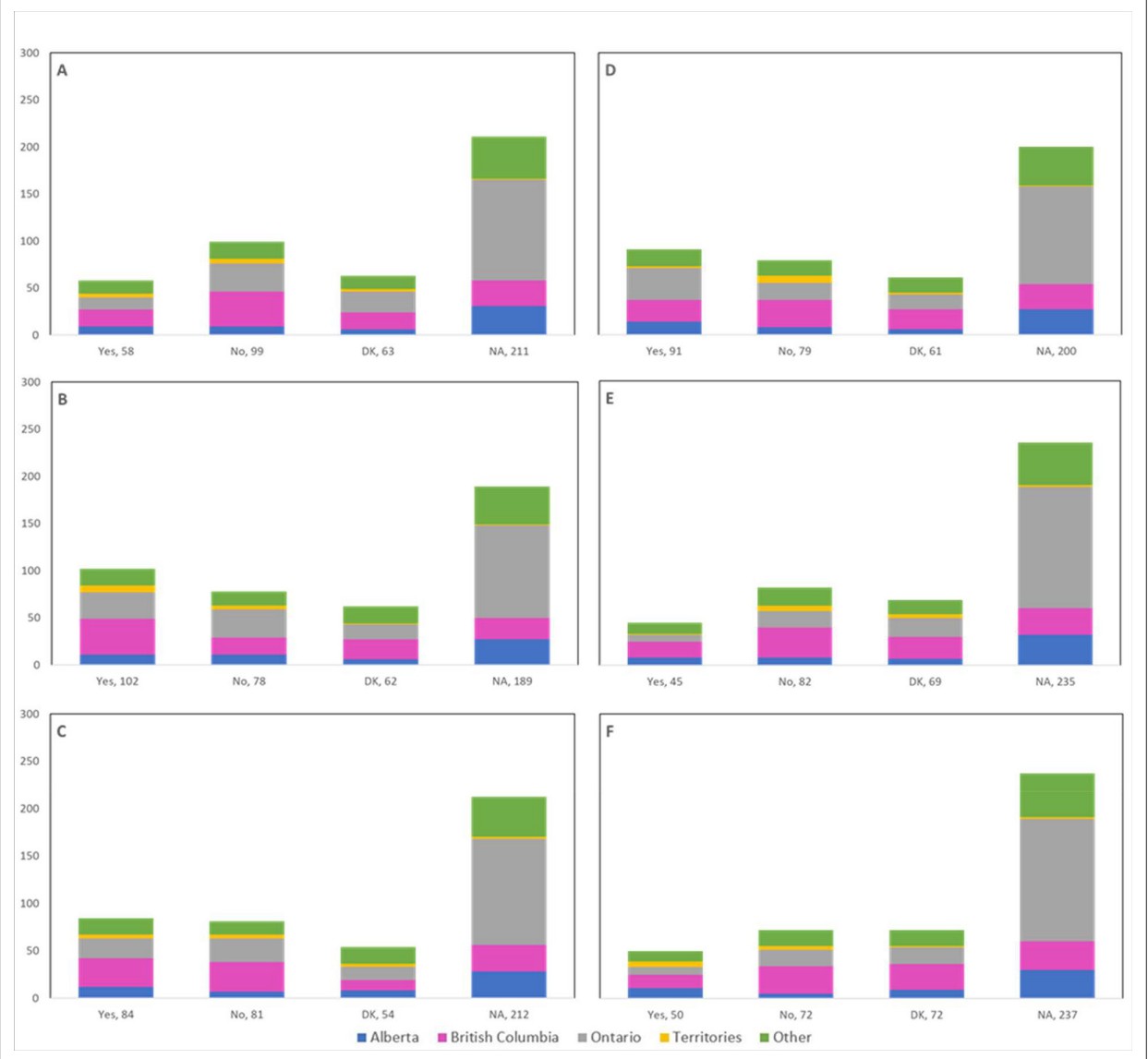

**Figure 10.** Cancellations or postponements of treatment procedures by province (n=431). Number of cancellations or postponements of (**A**) cold knife conization, (**B**) other excisional (e.g., LEEP), (**C**) ablative procedures, (**D**) hysterectomy, (**E**) chemotherapy, and (**F**) radiation are shown by province. Answers include the responses for question 39 by question 2. Territories include Northwest Territories, Nunavut, and Yukon. Other provinces include Manitoba, New Brunswick, Newfoundland and Labrador, Nova Scotia, Prince Edward Island, Quebec, and Saskatchewan (and one respondent who preferred not to say). DK: Don't know; NA: Not applicable to my practice.

procedures, in comparison to the pre-COVID-19 era. Most of these respondents were from Ontario, practicing in primary care settings at community hospitals (*Figure 13*). Notably, 32.3% mentioned an increase in the frequency of patients presenting with worsened symptoms, and 16.6% reported that 25–49% of patients had been diagnosed with more advanced cytological abnormalities and/or lesions confirmed by histology, compared to the pre-COVID-19 era (*Table 5*).

## Answers to open-ended questions

*Table 6* presents a categorization of the open-ended feedback provided by respondents. Several topics were discerned among the diverse raw responses for each open question. Around 40% of respondents mentioned that the pandemic would facilitate HPV self-sampling and is a favorable approach to implement in cervical cancer screening. Several challenges were described including operational, implementation, and evaluation considerations as well as healthcare system considerations.

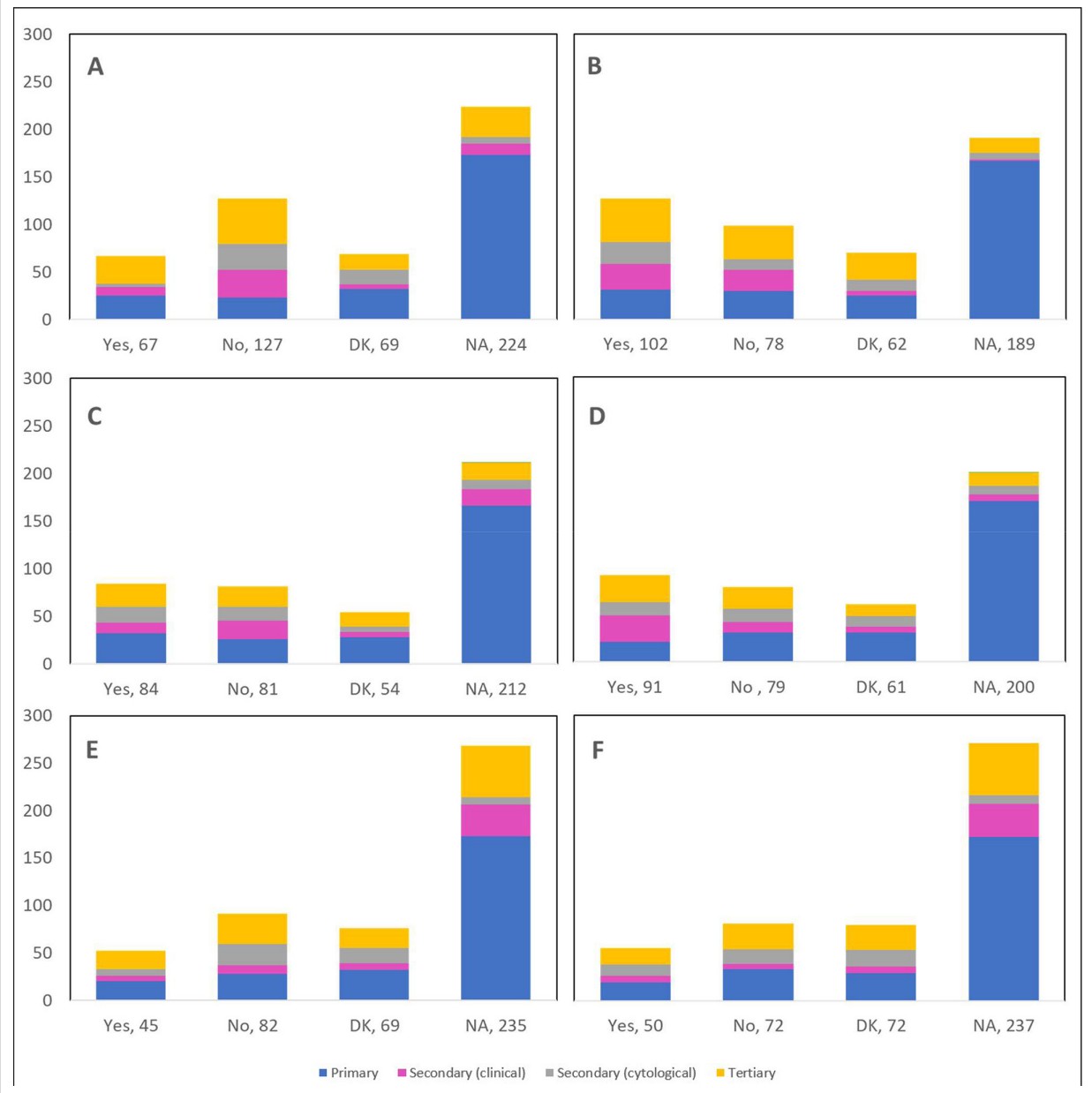

**Figure 11.** Cancellations or postponements of treatment procedures by profession (n=431). Number of cancellations or postponements of (**A**) cold knife conization, (**B**) other excisional (e.g., LEEP), (**C**) ablative procedures, (**D**) hysterectomy, (**E**) chemotherapy, and (**F**) radiation are shown by profession. Answers include the responses for question 39 by question 4. Primary includes general practitioners/family physicians, nurse practitioners/registered nurses, physician assistants, and a manager of a community health center; secondary (clinical) includes colposcopists and colposcopy registered nurses/registered practical nurses; secondary (cytological) includes cytopathologists/technologists and pathologists; tertiary includes gynecologists/ obstetrician-gynecologists, gynecology oncologists, and gynecology nurses. Frequency count exceeded total number of respondents as some reported multiple professions. DK: Don't know; NA: Not applicable to my practice.

Of the 206 responses to Q22, 30 survey respondents stated that they were not familiar with HPV self-sampling, whether it be with the mechanism or validity of the test. The vast majority of those who were not familiar with HPV self-sampling were primary care providers (90.0%), and the largest proportion were in Ontario (43.3%) and worked in private clinics (56.7%) (*Figure 14*). Similarly, 60 of the 197 respondents to Q23 explained that they were not familiar enough with HPV self-sampling to express a favorable or unfavorable opinion about its implementation as an alternative screening method. Of those, most were in Ontario (56.7%), were primary care providers (75.0%), and worked in private clinics (40.0%) (*Figure 15*). Not surprisingly, of those who responded that they were 'maybe' in favor

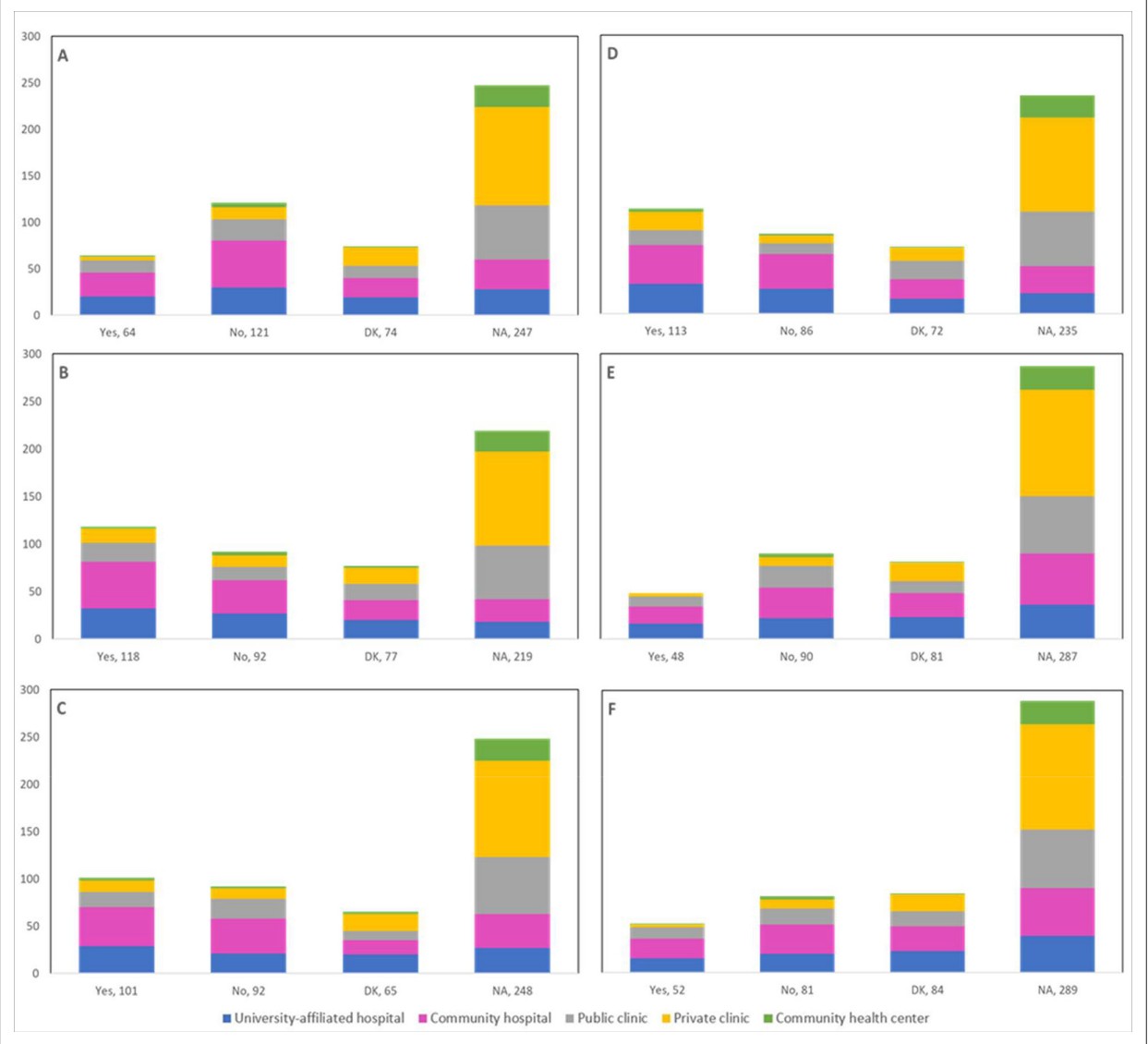

**Figure 12.** Cancellations or postponements of treatment procedures by place of practice (n=431). Number of cancellations or postponements of (**A**) cold knife conization, (**B**) other excisional (e.g., LEEP), (**C**) ablative procedures, (**D**) hysterectomy, (**E**) chemotherapy, and (**F**) radiation are shown by place of practice. Answers include the responses for question 39 by question 5. Frequency count exceeded total number of respondents as some reported multiple places of practice. DK: Don't know; NA: Not applicable to my practice.

of implementing HPV self-sampling as an alternative screening procedure, 47.1% (49/104) reported that they were not familiar enough with this screening modality to express their view. Respondents identified additional interactions deemed appropriate to convert to telemedicine, such as counselling services, follow-up with the patient, discussion of treatment options, and research-related activities. A substantial portion (36.2%) stated that no measures were implemented by their practice/institution to catch up with cancellations, postponements, and ongoing delays. Almost one-quarter of respondents to Q52 (24.7%) were forced to interrupt the services at their practice or institution for between 2 and 4 months due to the pandemic, and 13.1% reported interruptions of over 6 months. Many of those who did not experience interruptions to their practice (11.2%) described severely reduced services, deferral of patients with lower risk or lower grade disease, and use of telemedicine. Respondents had different interpretations of Q60; whereas a few reported the continuation of regular practice (10.3%) and use of personal protective equipment (4.4%) to screen COVID-19-positive patients, most (54.4%) reported that appointments were deferred until after the patient's isolation period. When asked about

**Table 3.** Adoption of telemedicine (n=429*).

| Question number and content | Categories | n (%) |
|---|---|---|
| Q43 Adoption of **telemedicine** to **communicate with patients** | Yes, with all patients | 179 (41.7) |
| | Yes, with low-risk patients only | 205 (47.8) |
| | No | 23 (5.4) |
| | *Don't know* | *9 (2.1)* |
| | *Not applicable to my practice* | *13 (3.0)* |
| Q44 Percentage of patients called (audio/video) for **distance consultation** | 0% | 17 (4.0) |
| | 1–24% | 83 (19.4) |
| | 25–49% | 115 (26.8) |
| | 50–74% | 105 (24.5) |
| | ≥75% | 83 (19.4) |
| | *Don't know* | *5 (1.2)* |
| | *Not applicable to my practice* | *21 (4.9)* |
| Q45 Percentage of patients called (audio/video) for **follow-up** to a cervical cancer screening procedure | 0% | 46 (10.7) |
| | 1–24% | 103 (24.0) |
| | 25–49% | 85 (19.8) |
| | 50–74% | 83 (19.3) |
| | ≥75% | 68 (15.9) |
| | *Don't know* | *18 (4.2)* |
| | *Not applicable to my practice* | *26 (6.1)* |
| Q46 **Virtual consultations compensated** by jurisdictional public health insurance system | Yes | 312 (72.7) |
| | No | 43 (10.0) |
| | *Don't know* | *44 (10.3)* |
| | *Not applicable to my practice* | *30 (7.0)* |
| Q47 Appropriate **interactions to convert to telemedicine**[†] | Health and medical history reporting | 283 (66.0) |
| | Test results reporting | 352 (82.1) |
| | Consent forms completion prior to in-person procedures | 222 (51.7) |
| | Post-procedure follow-up | 219 (42.9) |
| | In-person appointment planning/scheduling | 169 (33.1) |
| | Other | 5 (1.2) |

*Eighty-one respondents did not answer; the total number of complete responses was used as the denominator.
[†]Frequency count exceeded number of respondents (429) as some selected more than one answer.

which cervical cancer screening guidelines the respondent's practice/institution has been following, 63.5% answered governmental and 17.3% answered professional association/society.

## Discussion

We describe in the current paper findings from an online Canada-wide survey of healthcare professionals, capturing their opinions, perceptions, and work experience in relation to the impact of the disruptions in routine cervical cancer screening and resulting restrictions on colposcopy services during the early period of the COVID-19 pandemic. We report descriptive results on all survey questions, with explicit annotation to each question number and reference to the formulated question in

**Table 4.** Over-screening and under-screening in the pre-coronavirus disease 2019 (pre-COVID-19) era (n=427*).

| Question number and content† | Categories | n (%) |
|---|---|---|
| Q48 Prevalence of **over**-screening/over-diagnosis/over-treatment of cervical lesions pre-COVID-19 | Yes, over-screening | 68 (15.9) |
| | Yes, over-diagnosis | 86 (20.1) |
| | Yes, over-treatment | 36 (8.5) |
| | No | 224 (52.5) |
| | *Don't know* | *37 (8.7)* |
| Q49 **Current delays/cancellations** of screening/management procedures have had a **positive impact** by reducing unnecessary screening/diagnosis/treatment | Yes, over-screening | 65 (15.2) |
| | Yes, over-diagnosis | 88 (20.6) |
| | Yes, over-treatment | 45 (10.5) |
| | No | 188 (44.0) |
| | *Don't know* | *66 (15.5)* |
| Q50 Prevalence of **under**-screening/under-diagnosis/under-treatment of cervical lesions pre-COVID-19 | Yes, under-screening | 161 (37.7) |
| | Yes, under-diagnosis | 127 (29.7) |
| | Yes, under-treatment | 62 (14.5) |
| | No | 109 (25.5) |
| | *Don't know* | *54 (12.6)* |
| Q51 **Current delays/cancellations** of screening/management procedures have had a **negative impact** by reducing necessary screening/diagnosis/treatment | Yes, under-screening | 204 (47.8) |
| | Yes, under-diagnosis | 207 (48.4) |
| | Yes, under-treatment | 129 (25.3) |
| | No | 48 (9.4) |
| | *Don't know* | *50 (11.7)* |

*Eighty-three respondents did not answer; the total number of complete responses was used as the denominator.
†Frequency count exceeded number of respondents (427) as some selected more than one answer.

the appended survey instrument. Overall responses were reflective of the decline in cervical cancer screening and the challenges healthcare professionals faced when the pandemic was declared.

The pandemic's negative consequences and collateral damage have been consistently observed on an international level in relation to the pauses/postponed cancer screening programs. Screening rates in the United States had dropped 35% below averages of previous years (2017–2019) in the period of January to June 2020, with an estimated 40,000 missing screening tests through March to June (*Cox et al., 2021*). A population-based study in the United States reported a 46.4% decrease in the weekly number of newly identified patients with breast, colorectal, lung, pancreatic, gastric, and esophageal cancers during March to April 2020 (*Kaufman et al., 2021*). A modeling study in the United Kingdom found that a 2-month delay in the 2-week-wait investigatory referrals for suspected cancer can lead to an estimated loss of 0–0.7 life-years per patient (*Sud et al., 2020*). In the Netherlands, a notable decrease in cancer diagnoses between February 24, 2020, and April 12, 2020, was also reported compared with the period before the COVID-19 outbreak (*Dinmohamed et al., 2020*). Decreases were observed in another study conducted in Hong Kong, where weekly colorectal cancer diagnoses had fallen by 54% during the pandemic (*Lui et al., 2020*).

Particularly for cervical cancer, a 2-month screening lockdown between March 12, and May 8, 2020, in Slovenia resulted in a rapid decline in screening (−92%), follow-up (−70%), and HPV triage tests (−68%), in addition to invasive diagnostic (−47%) and treatment (−15%) of cervical lesions, compared to a 3-year average of years 2017–2019 (*Ivanuš et al., 2021*). An 83% decrease in the number of Pap tests was seen in Manitoba, Canada, in April 2020, most likely related to limited accessibility to primary healthcare providers (*Decker et al., 2022*). During the first 6 months of the pandemic

**Table 5.** Resumption of in-person practice (n=421*).

| Question number and content | Categories | n (%) |
|---|---|---|
| Q53 Practice/institution **caught up with cancellations/ postponements** | Yes | 190 (45.1) |
| | No | 147 (34.9) |
| | *Don't know* | *58 (13.8)* |
| | *Not applicable to my practice* | *26 (6.2)* |
| Q54 **Measures** implemented to **catch up** with cancellations/postponements† | Allow longer workdays and/or working on weekends | 91 (21.6) |
| | Increase availability of OR for treatment procedures | 91 (21.6) |
| | Convert OR procedures, if possible, to take place in clinics | 74 (17.6) |
| | Increase availability to labs for processing test samples | 48 (11.4) |
| | Other | 51 (11.9) |
| | *Don't know* | *41 (9.7)* |
| | *Not applicable to my practice* | *92 (21.9)* |
| Q55 Percentage of **cancellations/postponements** currently **caught up with** | 0% | 5 (1.2) |
| | 1–24% | 60 (14.3) |
| | 25–49% | 98 (23.3) |
| | 50–74% | 72 (17.1) |
| | ≥75% | 88 (20.9) |
| | *Don't know* | *55 (13.1)* |
| | *Not applicable to my practice* | *43 (10.2)* |
| Q56 Patients **attending routine screening at equivalent capacity** to pre-COVID-19 era | Yes | 132 (31.4) |
| | No | 216 (51.3) |
| | *Don't know* | *52 (12.4)* |
| | *Not applicable to my practice* | *21 (5.0)* |
| Q57 Percentage of patients attending **routine screening** compared **to pre-COVID-19** | 0% | 3 (0.7) |
| | 1–24% | 75 (17.8) |
| | 25–49% | 123 (29.2) |
| | 50–74% | 89 (21.1) |
| | ≥75% | 76 (18.1) |
| | *Don't know* | *35 (8.3)* |
| | *Not applicable to my practice* | *20 (4.8)* |
| Q58 **Increase in frequency** of patients with **worsening of symptoms** during screening | Yes | 136 (32.3) |
| | No | 216 (51.3) |
| | *Don't know* | *47 (11.2)* |
| | *Not applicable to my practice* | *22 (5.2)* |
| Q59 Percentage of patients diagnosed with **more advanced cytological abnormalities/lesions,** in comparison to pre-COVID-19 | 0% | 98 (23.3) |
| | 1–24% | 94 (22.3) |
| | 25–49% | 70 (16.6) |
| | 50–74% | 42 (10.0) |
| | ≥75% | 6 (1.4) |
| | *Don't know* | *95 (22.6)* |
| | *Not applicable to my practice* | *16 (3.8)* |

*Table 5 continued on next page*

Table 5 continued

| Question number and content | Categories | n (%) |
|---|---|---|
| | Yes | 113 (26.8) |
| | No | 222 (52.7) |
| Q60 **Screening** patients (**with COVID-19**) for cervical cancer | *Don't know* | *58 (13.8)* |
| | *Not applicable to my practice* | *28 (6.7)* |

*Eighty-nine respondents did not answer; the total number of complete responses was used as the denominator.
†Frequency count exceeded number of respondents (421) as some selected more than one answer.

in Ontario, Canada, there was a decrease in the monthly average number of Pap tests (−63.8%), colposcopies (−39.7%), and treatments (−31.1%), compared with the corresponding months in 2019 (*Meggetto et al., 2021*).

Similarly, modeling data have consistently predicted an excess of cervical cancer cases and deaths caused by the scaling down of cervical screening and treatment services due to COVID-19 disruptions and resource constraints. Under a cytology-based screening model, suspensions of 6 and 24 months in the screening continuum in the United States were estimated to yield an additional 5–7 and an additional 25–27 cases of cervical cancer, respectively, by 2027. The numbers of increased cases were greater for women previously screened with cytology compared with co-testing (cytology plus HPV testing) (*Burger et al., 2021*). A 25.7% decrease in diagnosis of low stage cervical cancer was observed in the North of England during the pandemic compared to 2019 (*Davies et al., 2022*). The authors estimated a total of 919 cases of cervical cancer will by 2023, compared to 233 cases pre-COVID (May to October 2019), caused by a lack of diagnosis of established cases and an excess of cases caused by lack of screening. In India, delays in diagnoses and treatment were estimated to result in a 2.5% (n=795) to 3.8% (n=2160) lifetime increase in the deaths caused by cervical cancer, compared to no delays (*Gupta et al., 2021*).

Our survey results pointed to a potential for the use of self-collected samples for HPV-based screening and the need for adaptability. The World Health Organization's call to eliminate cervical cancer (*World Health Organization, 2020*) has motivated efforts across the globe to scale up screening services and introduce a paradigm shift in cervical screening by implementing HPV-based

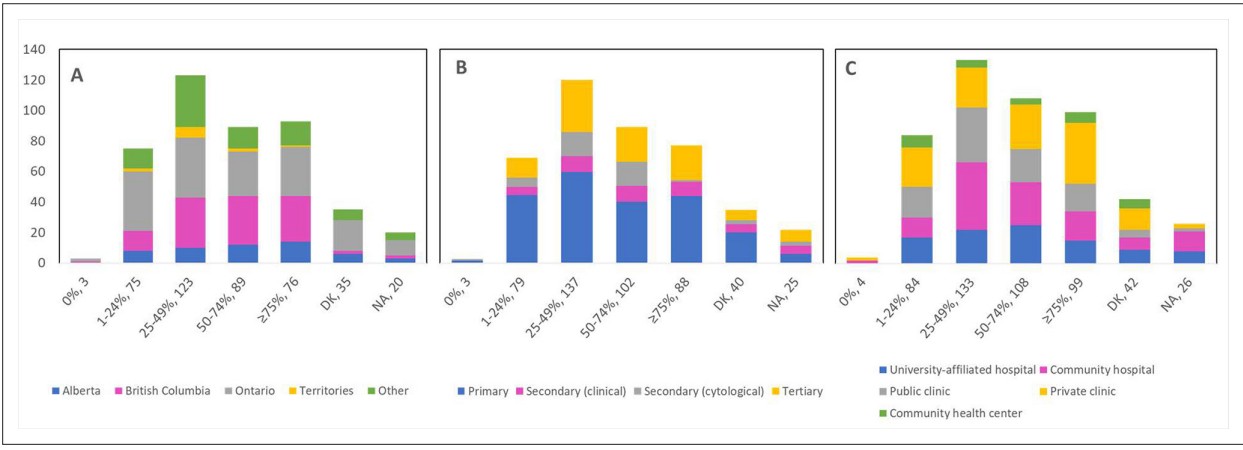

**Figure 13.** Percentage of patients attending routine screening compared to pre-coronavirus disease 2019 (pre-COVID-19) by province, profession, and place of practice (n=421). Proportions are shown by (**A**) province, (**B**) profession, and (**C**) place of practice. Answers include responses for question 57 by questions 2 (province), 4 (profession), and 5 (place of practice). Panel A: Territories include Northwest Territories, Nunavut, and Yukon. Other provinces include Manitoba, New Brunswick, Newfoundland and Labrador, Nova Scotia, Prince Edward Island, Quebec, and Saskatchewan (and one respondent who preferred not to say). Panel B: Primary includes general practitioners/family physicians, nurse practitioners/registered nurses, physician assistants, and a manager of a community health center; secondary (clinical) includes colposcopists and colposcopy registered nurses/registered practical nurses; secondary (cytological) includes cytopathologists/technologists and pathologists; tertiary includes gynecologists/obstetrician-gynecologists, gynecology oncologists, and gynecology nurses. Panels B and C: Frequency count exceeded total number of respondents as some reported multiple places of practice. DK: Don't know; NA: Not applicable to my practice.

**Table 6.** Content analysis of open-ended questions.

| Question number and content (number of responses) | Opinions and perspectives | n (%) |
|---|---|---|
| | Favorable approach | 85 (41.3) |
| | Not favorable | 18 (8.7) |
| | Challenges faced* | 47 (22.8) |
| | Not familiar with HPV self-sampling[†] | 30 (14.6) |
| | *'No comment'* written | 6 *(2.9)* |
| Q22 **COVID-19** to encourage/facilitate/accelerate **implementation of HPV self-sampling** in cervical cancer screening programs, briefly justify your answer (n=206) | *Don't know* | 12 *(5.8)* |
| | *Unclear answer* | 8 *(3.9)* |
| | Favorable approach | 80 (40.1) |
| | Not favorable | 12 (6.1) |
| | Challenges described* | 29 (14.7) |
| | Not familiar with HPV self-sampling[†] | 60 (30.5) |
| | *'No comment'* written | 2 (1.0) |
| Q23 **In favor** of implementing **HPV self-sampling** as alternative screening method in practice, briefly justify your answer (n=197) | *Don't know* | 4 *(2.0)* |
| | *Unclear answer* | 10 *(5.1)* |
| | All of the above, but not in all cases | 1 (20.0) |
| | Counselling and family meetings | 1 (20.0) |
| | Research-related activities | 1 (20.0) |
| Q47 Appropriate **interactions to convert to telemedicine,** other (n=5) | Follow-up any issues | 1 (20.0) |
| | Discuss treatment options | 1 (20.0) |
| | No interruption | 47 (11.2) |
| | <1 month | 41 (9.7) |
| | 1 month to <2 months | 51 (12.1) |
| | 2 months to <4 months | 104 (24.7) |
| | 4 months to <6 months | 47 (11.2) |
| | >6 months | 55 (13.1) |
| | *Don't know* | 5 *(1.2)* |
| | *Not applicable to my practice* | 3 *(0.7)* |
| Q52 **Duration of service interruption** in practice/institution due to pandemic, before resumption (n=421) | *Unclear answer* | 68 *(16.2)* |
| | Increased screening capacity (clinic space and staff) | 12 (20.7) |
| | Prioritizing patients | 2 (3.4) |
| | Adapting and enforcing screening criteria | 3 (5.2) |
| | Allowing in-person screening | 3 (5.2) |
| | Contacting and rebooking patients | 6 (10.3) |
| | Telemedicine | 4 (6.9) |
| | Screening continued during COVID-19 | 2 (3.4) |
| Q54 **Measures** implemented to **catch up** with cancellations/postponements, other (n=58) | None | 21 (36.2) |
| | *Unclear answer* | 5 *(8.6)* |

*Table 6 continued on next page*

*Table 6 continued*

| Question number and content (number of responses) | Opinions and perspectives | n (%) |
|---|---|---|
| | Only those who are asymptomatic | 1 (1.5) |
| | COVID-19 screening pre-appointment | 12 (17.6) |
| | Use of PPE | 3 (4.4) |
| | Deferral | 37 (54.4) |
| | Telemedicine | 1 (1.5) |
| | Regular practice | 7 (10.3) |
| Q60 **Screening** patients (**with COVID-19**) for cervical cancer, if yes, briefly describe the process of cervical cancer screening of COVID-19 patients' (n=68) ‡ | *Not applicable to my practice* | *3 (4.4)* |
| | *Unclear answer* | *6 (8.8)* |
| | Governmental | 268 (63.5) |
| | Local/institutional | 19 (4.5) |
| | Professional association/society | 73 (17.3) |
| | Cancer organization/society | 10 (2.4) |
| | None | 13 (3.1) |
| | *Don't know* | *11 (2.6)* |
| Q61 Which **cervical cancer screening guidelines** has your practice/institution been following (n=422)‡ | *Not applicable to my practice* | *4 (0.1)* |
| | *Unclear answer/acronym* | *40 (9.5)* |

*Include cost and whether it will be funded by the government; the need to be added to the guidelines and endorsed by government and professionals along with having a well-designed program that helps with patient compliance and the need for professionals to be well educated on the subject; implementation challenges (including delays due to the pandemic, burnout, lack of available healthcare spending, lack of appropriate healthcare infrastructure, lack of prioritization of women's health); patient education (awareness, proper technique given with clear simple instructions); and logistics (material currently not available or test not routinely offered, should kits be mailed to participants).

†Respondents were either not familiar with the test itself, with whether the test is available, or with the test's validity (in terms of its sensitivity and specificity).

‡Frequency count exceeded number of respondents (68 respondents for Q60 and 422 respondents for Q61) as some provided more than one answer.

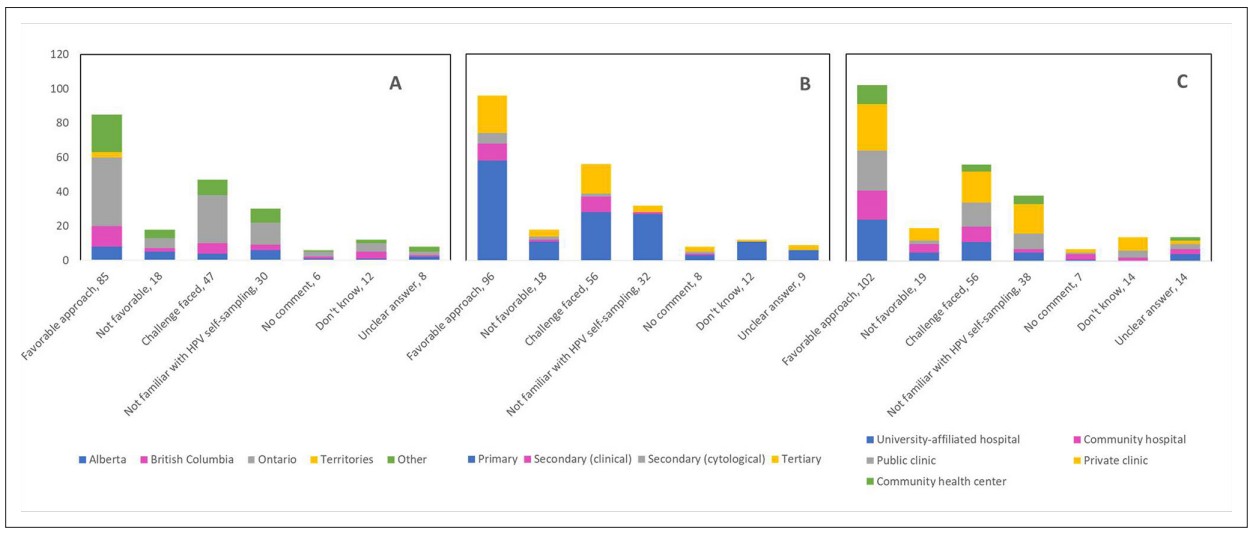

**Figure 14.** Responses to open-ended Q22 by province, profession, and place of practice (n=206). Opinions and perspectives are shown by (**A**) province, (**B**) profession, and (**C**) place of practice. Answers include responses for question 22 (coronavirus disease 2019 [COVID-19] to encourage/facilitate/accelerate implementation of human papillomavirus [HPV] self-sampling) by questions 2 (province), 4 (profession), and 5 (place of practice). Panel A: Territories include Northwest Territories, Nunavut, and Yukon. Other provinces include Manitoba, New Brunswick, Newfoundland and Labrador, Nova Scotia, Prince Edward Island, Quebec, and Saskatchewan (and one respondent who preferred not to say). Panel B: Primary includes general practitioners/family physicians, nurse practitioners/registered nurses, physician assistants, and a manager of a community center; secondary (clinical) includes colposcopists and colposcopy registered nurses/registered practical nurses; secondary (cytological) includes cytopathologists/technologists and pathologists; tertiary includes gynecologists/obstetrician-gynecologists, gynecology oncologists, and gynecology nurses. Panels B and C: Frequency count exceeded total number of respondents as some reported multiple professions and places of practice.

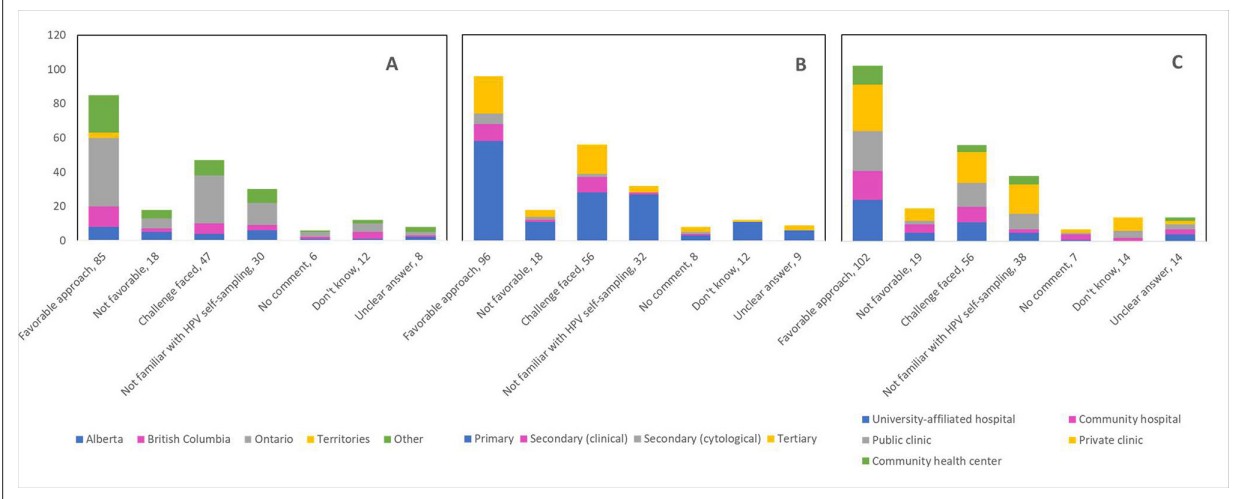

**Figure 15.** Responses to open-ended Q23 by province, profession, and place of practice (n=197). Opinions and perspectives are shown by (**A**) province, (**B**) profession, and (**C**) place of practice. Answers include responses for question 23 (in favor of implementing human papillomavirus [HPV] self-sampling as alternative screening method) by questions 2 (province), 4 (profession), and 5 (place of practice). Panel A: Territories include Northwest Territories, Nunavut, and Yukon. Other provinces include Manitoba, New Brunswick, Newfoundland and Labrador, Nova Scotia, Prince Edward Island, Quebec, and Saskatchewan (and one respondent who preferred not to say). Panel B: Primary includes general practitioners/family physicians, nurse practitioners/ registered nurses, physician assistants, and a manager of a community health center; secondary (clinical) includes colposcopists and colposcopy registered nurses/registered practical nurses; secondary (cytological) includes cytopathologists/technologists and pathologists; tertiary includes gynecologists/obstetrician-gynecologists, gynecology oncologists, and gynecology nurses. Panels B and C: Frequency count exceeded total number of respondents as some reported multiple professions and places of practice.

programs. HPV self-sampling addresses the challenges of COVID-19 (need for social distance and a possibility of at-home sample collection) and women's empowerment (samples collected by women themselves), thus offering a socially distanced approach that will substantially reduce the need for clinic appointments. Along the same lines, our results highlight the important role that telemedicine has played in mitigating the effects of delays in cervical cancer screening and follow-up and reducing the backlog faced upon the resumption of in-person practice.

There are some limitations to our purely descriptive study that need to be acknowledged. First, we were unable to fully reach out to general practitioners and family physicians who are mostly involved at the forefront of cervical cancer screening processes. The College of Family Physicians of Canada did not approve circulating the survey to their members to avoid inconveniencing them with external activities. However, we commissioned MDBriefCase to publicize the survey to community primary care providers. We also reached out to family physicians in academic medicine within university networks. Second, the non-response rate cannot be calculated due to the recruitment methods used (including advertisement of the survey on social media) and lack of information on the population of interest (number of professionals approached). However, our survey and findings should be considered on their merits; the target population was involved in the survey design to ensure the validity of coverage of the questions along the continuum of care in cervical cancer screening and treatment. Finally, 34% of surveys were excluded. Most often, this was because we did not include a screening question to ensure respondents were eligible to participate. Less often, there were multiple entries by the same respondent which could have resulted from the use of a snowball method (particularly via social media) and the lack of unique IP addresses. Our survey collection strategy did not enable valida-tion of the respondent's eligibility to participate in the survey. However, we used the answers to the demographic and open-ended questions to determine eligibility and legitimacy of the responses and verified that there were no duplicate surveys submitted. In all, most respondents carefully answered and provided candid views.

The strengths of this survey study are its Pan-Canadian scope and design querying five themes of the cervical cancer screening and treatment continuum, the widely publicized approach and endorsements by professional societies and organizations, and the participation of multiple health professional disciplines. Although the survey only provides a snapshot of the extent of the harms

to key cervical cancer screening and follow-up services at the beginning of the pandemic, further research and ongoing monitoring of health services utilization are needed to understand the full impact. However, our findings identified several key lessons for future response efforts and highlight the need for (1) properly formulated recommendations and strategies that would help mitigate the negative outcomes of the pandemic, (2) development of potential recovery strategies (i.e., risk-based triage systems as well as awareness campaigns on the importance and value of cervical cancer screening) for resuming routine cervical cancer screening, and (3) help building resilience in screening processes. Our survey provides evidence to support the implementation of HPV-based programs and the use of telemedicine to continue cervical cancer screening, treatment, and follow-ups and reduce backlogs while mitigating inconveniences to both patients and healthcare professionals. In addition, insights from the survey could inform epidemiological modeling studies of the long-term effects of the interruptions and delays in screening activities on cervical cancer morbidity and mortality.

## Acknowledgements

The authors are grateful to all the societies that assisted with survey distribution: SCC, GOC, CAP, and SOGC. We are also grateful to Dr. Wilson H Miller, Director of the COVID-19 and Cancer Program (McGill University) for the insights and support. We thank Dr. Helene Trottier (University of Montreal) for verification of the French translation of the survey questions. Finally, we express our deepest gratitude to the respondents for taking precious time out of their work schedule to share their valuable perspectives.

## Additional information

### Group author details

**Survey Study Group**
**Aisha Lofters; Amanda Selk; Barbara Bodmer; Celine Bouchard; Meg McLachlin; Fady W Mansour; Gina Ogilvie; James Bentley; Jennifer Blake; Jessica Papillon Smith; Joan Murphy; Kathleen Decker; Lorraine Elit; Lucy Gilbert; Maire A Duggan; Marette Lee; Marie-Helene Mayrand; Paul Bessette; Rachel Kupets; Shannon Salvador; Suzie Lau; Walter H Gotlieb**

### Competing interests

Rami Ali: received an MSc. stipend from the Gerald Bronfman Department of Oncology, McGill University. Eduardo L Franco: Senior editor, *eLife*. The other authors declare that no competing interests exist.

### Funding

| Funder | Grant reference number | Author |
|---|---|---|
| Canadian Institutes of Health Research | VR5-172666 | Eduardo L Franco |

The funders had no role in study design, data collection and interpretation, or the decision to submit the work for publication.

### Author contributions

Mariam El-Zein, Conceptualization, Resources, Data curation, Formal analysis, Supervision, Funding acquisition, Investigation, Methodology, Writing – original draft, Project administration, Writing – review and editing; Rami Ali, Eliya Farah, Data curation, Formal analysis, Methodology, Project administration, Writing – review and editing; Sarah Botting-Provost, Data curation, Formal analysis, Validation, Visualization, Methodology, Project administration, Writing – review and editing; Eduardo L Franco, Conceptualization, Resources, Supervision, Funding acquisition, Project administration, Writing – review and editing; Survey Study Group, Provided feedback on the relevance, appropriateness, and clarity of theme-related survey questions, and on questionnaire length

## Author ORCIDs
Mariam El-Zein (iD) https://orcid.org/0000-0002-5190-0370
Sarah Botting-Provost (iD) http://orcid.org/0000-0001-7510-0258
Eduardo L Franco (iD) http://orcid.org/0000-0002-4409-8084

## Ethics

The Faculty of Medicine Institutional Review Board (IRB) of McGill University granted ethical approval for this work on October 27, 2020 (IRB Internal Study Number: A10-B84-20A). All necessary participant consent has been obtained and the appropriate institutional forms have been archived.

## Decision letter and Author response
Decision letter https://doi.org/10.7554/eLife.83764.sa1
Author response https://doi.org/10.7554/eLife.83764.sa2

---

# Additional files

## Supplementary files
• Supplementary file 1. Pan-Canadian survey on the impact of the coronavirus disease 2019 (COVID-19) pandemic on cervical cancer screening and management. Hard copy of the online survey instrument.

• Supplementary file 2. Tabulation of survey questions. Includes frequency distribution of Q11-13 (2a), Q14-16 (2b), Q17-19 (2c), Q20-21 (2d), Q22-23 (2e), Q24-29 (2f), Q30-35 (2g), Q36-37 (2h), and Q38-Q42 (2i).

• MDAR checklist

## Data availability

The survey instrument appears in *Supplementary file 1*. Code files and datasets corresponding to analyses and descriptive figures included in the manuscript and supplement are available online at Borealis, the Canadian Dataverse Repository. Source data for all figures, both embedded and supplementary, are available at the Dataverse link (https://doi.org/10.5683/SP3/8MVU6L) with filename ' CxCaSurvey_Data.tab'.

The following dataset was generated:

| Author(s) | Year | Dataset title | Dataset URL | Database and Identifier |
|-----------|------|---------------|-------------|-------------------------|
| Botting-Provost S, El-Zein M | 2022 | Pan-Canadian Survey on the Impact of the COVID-19 Pandemic on Cervical Cancer Screening and Management | https://doi.org/10.5683/SP3/8MVU6L | Borealis, 10.5683/SP3/8MVU6L |

---

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
