## [Editor Report]

This study explored practitioners' assessments of the impact of the pandemic on cervical cancer screening and follow-up. This is a very important topic that could continue to have implications for how this screening process is delivered now, after the pandemic.

---

## [Decision Letter]

**Decision letter after peer review:**

Thank you for submitting your article "Pan-Canadian survey on the impact of the COVID-19 pandemic on cervical cancer screening and management: cross-sectional descriptive study among healthcare professionals" for consideration by *eLife*. Your article has been reviewed by 2 peer reviewers, and the evaluation has been overseen by a Reviewing Editor and Diane Harper as the Senior Editor. The reviewers have opted to remain anonymous.

Essential revisions:

1. Please describe the limitations of your methodology.

2. Please temper your conclusions to match the rigor of your methodology.

*Reviewer #1 (Recommendations for the authors):*

I recommend a clearer focus on the main findings and a more humble discussion, pointing out the weaknesses of the study. I recommend an assessment of the survey coverage and a discussion of the implication of low coverage.

*Reviewer #2 (Recommendations for the authors):*

This study relies mostly on descriptive statistics and open-ended questions that provide details about what CC screening and treatment procedures were delayed. It is unclear how the reader would use the results to affect current or future practice. The analytic approach would have been strengthened with correlational analyses to better understand what factors were associated with the cancellations or delays of CC screenings and treatment in the context of the pandemic, such as how the various types of clinical settings were associated with particular procedures or length of delays. Such information can inform policy on how local and national governments can support the various clinical practices to build the capacity to meet patient services demands and avoid future negative disease outcomes when public health or environmental crisis occurs.

---

## [Author Response]

Essential revisions:1. Please describe the limitations of your methodology.2. Please temper your conclusions to match the rigor of your methodology.

We appreciate the positive feedback and are thankful for the opportunity to submit a revised version of our manuscript. We have made several modifications (detailed below), where applicable, to mainly address the listed concerns and recognize the limitations of our methodology. The comments were very helpful in improving our manuscript.

Reviewer #1 (Recommendations for the authors):I recommend a clearer focus on the main findings and a more humble discussion, pointing out the weaknesses of the study. I recommend an assessment of the survey coverage and a discussion of the implication of low coverage.

We appreciate all the suggestions, and we now address all concerns.

Reviewer #2 (Recommendations for the authors):This study relies mostly on descriptive statistics and open-ended questions that provide details about what CC screening and treatment procedures were delayed. It is unclear how the reader would use the results to affect current or future practice. The analytic approach would have been strengthened with correlational analyses to better understand what factors were associated with the cancellations or delays of CC screenings and treatment in the context of the pandemic, such as how the various types of clinical settings were associated with particular procedures or length of delays. Such information can inform policy on how local and national governments can support the various clinical practices to build the capacity to meet patient services demands and avoid future negative disease outcomes when public health or environmental crisis occurs.

While we did provide results for cross-tabulations between province, profession, and place of practice and cancellations and postponement of screening, treatment, and follow-up procedures in Figures 2, and 5 though 9, as well as the length of screening appointment deferrals by respondents’ characteristics in Figure 3, we welcome the opportunity to add additional details on the length of deferrals by place of practice. We have conducted additional cross-tabulations and have added the following text to the results:

Theme 1: Screening practice

“Of the respondents who experienced Pap test deferral periods of 2 months or longer, 38% (100/26) worked in private clinics. Those who practiced in community hospitals reported deferral periods of 2 months or more for HPV test (36.8% [42/114]) and HPV/Pap co-test (42.7% [53/124]) more frequently than other places of practice.”

Changes made: Page 8, Lines 232-235.

“Professionals from community hospitals saw longer deferral periods for postponed colposcopy appointments than those from other settings, accounting for 40.9% (38/93) of deferrals of 2 months or longer, whereas respondents from community health centers accounted for 34.9% (38/109) of deferrals of 2 months or longer of follow-up appointments.”

Changes made: Page 9, Lines 264-268.

Theme 2: Treatment of pre-cancerous lesions and cancer

“Community hospitals accounted for almost half of deferrals of 2 months or longer of cold knife conisation procedures (48.9% [43/88]), other excisional procedures (47.6% [39/82]), ablative procedures (46.7% [43/92]), hysterectomies (49.0% [47/96]), chemotherapy (46.8% [36/77]), and radiation (48.6% [35/72]).”

Changes made: Page 10, Lines 282-285.